

# InSAR-based characterization of rock glacier movement in the Uinta Mountains, Utah, USA

George Brencher[1], Alexander L. Handwerger[2,3], Jeffrey S. Munroe[1]

[1]Geology Department, Middlebury College, Middlebury, 05753, USA
[2]Joint Institute for Regional Earth System Science and Engineering, University of California, Los Angeles, 90095, USA
[3]Jet Propulsion Laboratory, California Institute of Technology, Pasadena, 91109, USA

*Correspondence to*: George Brencher (qbrencher@gmail.com)

**Abstract.** Rock glaciers are a prominent component of many alpine landscapes and constitute a significant water resource in some arid mountain environments. Here, we employ satellite-based interferometric synthetic aperture radar (InSAR) to

identify and monitor active rock glaciers in the Uinta Mountains (Utah, USA), an area of ~10,000 km². We used mean velocity maps to generate an inventory for the Uinta Mountains containing 255 active rock glaciers. Active rock glaciers are 10.8 ha in area on average, and located at a mean elevation of 3290 m, where mean annual air temperature is 0.12 ˚C. The mean line-of-sight (LOS) velocity for the inventory is 2.52 cm/yr, but individual rock glaciers have LOS velocities ranging from 0.88 to 5.26 cm/yr. To search for relationships with climatic drivers, we investigate the time-dependent motion of three

rock glaciers over the summers of 2016–2019. Time series analysis suggests that rock glacier motion has a significant seasonal component, with motion that is more than 5 times faster during the late summer compared to rest of the year. Rock glacier velocities also appear to be correlated with the snow-water equivalent of the previous winter's snowpack. These results demonstrate the ability to use satellite InSAR to monitor rock glaciers over large areas and provide insight into the environmental factors that control their kinematics.

## 1 Introduction

Rock glaciers are perennially frozen bodies of ice and rock debris that creep downslope due to deformation of their internal ice-rock mixture (Fig. 1) (Wahrhaftig and Cox, 1959; Barsch, 1996). They play an important role in alpine hydrology and landscape evolution, principally through the release of seasonal meltwater and the continuous downslope transport of coarse material (Azócar and Brenning, 2010; Frauenfelder and Kääb, 2000). They also constitute a significant water resource in arid

regions (Schaffer et al., 2019), and their importance as a water source is likely to increase with ongoing climate change (Jones et al., 2018a). Understanding how rock glaciers respond to changes in climate is therefore a necessary part of predicting the evolution of cold mountain environments.

Active rock glaciers contain internal ice and typically flow downslope at rates on the order of 0.1–2 m/yr (Delaloye et al.,

2010). Controls on rock glacier kinematics are numerous, and include time-dependent water and talus delivery, internal rock



glacier structure, bedrock geometry, and short and long-term changes in ground temperature (Haeberli et al., 2006; Kääb et al., 2007; Ikeda et al., 2008; Delaloye et al., 2010; Müller et al., 2016; Kenner et al., 2017). As with ice glaciers, tectonic faults and landslides, liquid water, and pore-water pressure are also important drivers of short-term rock glacier motion (Ikeda et al., 2008; Moore, 2014; Kenner et al., 2017; Eriksen et al., 2016; Cicoira, 2019; Fey and Krainer, 2020).

Rock glacier velocity has been shown to vary over temporal scales ranging from multi-year to hourly (Delaloye et al., 2010; Kenner et al., 2017). For instance, rock glaciers in the European Alps and Norway have undergone significant long-term acceleration in the past few decades attributed to warming permafrost temperatures (Delaloye et al., 2010; Kääb et al., 2007; Kaufmann and Ladstädter, 2007; Roer et al., 2005; Vonder Muehll et al., 2007; Ikeda et al., 2008; Kenner et al., 2017;
Eriksen et al., 2018; Fey and Krainer, 2020). On a multi-year time scale, air temperature is thought to primarily control variations in rock glacier velocity (Arenson et al., 2002; Haeberli et al., 2006), although differences in precipitation between years may also explain contrasts in rock glacier movement (Ikeda et al., 2008). Seasonal variations in rock glacier motion are also well documented (Haeberli, 1985; Kääb et al., 1997; Arenson et al., 2002) and typically manifest as a strong acceleration in the late spring coincident with melting of snow and ice, followed by a gradual deceleration in the late fall and
winter (Perruchoud and Delaloye, 2007; Liu et al., 2013). Recent work suggests that spring acceleration is driven by water infiltrating the shear zone at the base of the rock glacier, which increases pore-water pressure and reduces frictional strength (Kenner et al., 2017; Cocoira et al., 2019; Fey and Krainer, 2020). Heavy precipitation and snowmelt events have been observed to coincide with short-term rock glacier acceleration (Wirz et al., 2016; Kenner et al., 2017).

Access to high-resolution remote sensing imagery and digital topography has improved our ability to inventory rock glaciers over large areas (Krainer and Ribis, 2012; Rangecroft et al., 2014; Falaschi et al., 2014; Roudnitska et al., 2016; Jones et al., 2018b; Brardinoni et al., 2019). Such datasets are important for assessing the past and current distributions of mountain permafrost, understanding landscape response to climate change following the last glacial maximum, and appraising the regional water storage potential of rock glaciers (Lieb, 1996; Boeckli et al., 2012; Schmid et al., 2015; Zasadni and Kłapyta,
2016; Azócar and Brenning, 2010; Jones et al., 2018a). However, most remote-sensing based inventories use optical images to identify rock glaciers based on morphology alone, and are therefore incapable of determining whether features are currently active (Munroe, 2018; Nicholson et al., 2009; Lilleøren and Etzelmüller, 2011; Kellerer-Pirklbauer et al., 2012; Rangecroft et al., 2014; Roudnitska et al., 2016). Optical imagery is also limited to daytime acquisitions, and snow, shadow, and clouds can obscure features on the ground, leading to undercounting of rock glaciers (Villarroel et al., 2018).


An alternate approach is Interferometric Synthetic Aperture Radar (InSAR), a remote sensing technique that is not affected by cloud cover, can operate day or night, can reveal deformation even in the presence of snow (Villarroel et al., 2018), and can detect motion on the millimeter scale. InSAR has been effectively used to create rock glacier inventories of varying extents and to study rock glacier kinematics (e.g. Nagler et al., 2002; Rignot et al., 2002; Kenyi and Kaufman, 2003

Lilleøren et al., 2013; Lui et al., 2013; Wang et al., 2017; Villarroel et al., 2018). InSAR does have limitations that are germane to the study of rock glaciers, including underestimation of true 3D velocities, complications arising from the fixed look direction of the satellite, atmospheric delay, poor correlation caused by changes in ground cover such as snow to snow-free conditions, and errors associated with features moving at velocities greater than the deformation threshold related to the radar wavelength. Many of these can be mitigated by careful study design, however, and at the scale of range-wide analysis,

significant patterns can still be identified.

Given the importance of rock glaciers as features of high mountain landscapes, their contribution to mountain hydrology, their ability to act as permafrost indicators, and the remaining open questions about climatic controls on their behavior, additional work is needed to develop the robust understanding of rock glacier dynamics necessary for predicting the response

of these features to future climate change (Eriksen et al., 2018). Here, we investigate rock glaciers in the Uinta Mountains (Utah, USA), an important headwater region for the Colorado River in the southwestern USA. Nearly 400 rock glaciers have been visually identified in these mountains, but the presence of ice has been indirectly investigated in only two, limiting understanding of slope processes, hydrology, and permafrost extent in this region (Munroe, 2018). We use satellite-based InSAR to identify active rock glaciers and to evaluate controls on their rates of motion over seasonal to multi-annual time

scales. Our work is the first to use satellite InSAR to investigate rock glacier motion in this mountain range.

### 2.1 Study area

The Uinta Mountains are an east-west trending mountain range extending over ~10,000 km$^2$ in northeastern Utah, USA (Fig. 2). The bedrock of the Uinta Mountains (hereafter, the "Uintas") is Precambrian quartzite, sandstone, and argillite uplifted during the Laramide Orogeny (70–80 Ma) (Bradley, 1936; Sears et al., 1982; Dehler, 2007). While the Uintas are not

currently glacierized, during the late Pleistocene, valley glaciers covered more than 2,000 km$^2$ of the range, creating well-developed glacial geomorphology (Munroe and Laabs, 2009).

Climate at higher elevations in the Uintas (> 3,000 m) is periglacial, with a mean annual air temperature (MAAT) from 1981–2010 ranging from -4.0 to 3.3 ℃ (Fig. 2b). Future climate conditions have not been modelled specifically for the

Uintas, but Atmosphere–Ocean General Circulation Models for the western U.S. parameterized with the A1B "business as usual" emissions scenario predict an increase in air temperature of about 1.5 ℃ by 2030 (IPCC, 2018). Average precipitation in the Uintas between 1981 and 2010 ranged from 45 to 107 cm (Fig. 2c). Most precipitation falls in the winter as snow, of which the largest quantities fall in the western part of the range. The Uintas are an important water resource in the state of Utah, supplying water to 17 counties for municipal, industrial, and agricultural purposes (Tingstad, 2010). Furthermore,

calculations indicate that 10–15% of the Colorado River flow at Lees Ferry, Arizona, is comprised of water from northeastern Utah (Tingstad, 2010). Flow of the Colorado River is expected to decline 10–30% from 2008 levels in the next



18 to 38 years based on streamflow modelling work, implying intense water deficits in the Colorado River system over the next century (Barnett and Pierce, 2008).

Rock glaciers and talus in the Uintas are estimated to store 0.14 km$^3$ of water, or ~10–35% of the range's annual runoff (Munroe, 2018). These rock glaciers could constitute an even more significant water resource in the coming years as climate change alters temperature and precipitation regimes in the western U.S., particularly during the late summer and autumn when there is less precipitation (MacDonald and Tingstad, 2007). Previous work identified 395 rock glaciers in the Uintas (Munroe, 2018). One of the primary goals of this study is to update this existing rock glacier inventory, which was based

solely rock glacier morphology, and to identify which of these features are currently active.

## 2.2 InSAR analysis

Copernicus Sentinel-1 radar data covering the Uinta Mountains were downloaded from the Alaska Satellite Foundation Vertex website (https://vertex.daac.asf.alaska.edu/). Interferometric wide swath (IW) mode, Single Look Complex (SLC) scenes from August 2016–October 2019 were selected based on whether Uinta rock glaciers appeared to be largely snow-

free in optical Planetlab imagery (https://www.planet.com/explorer/). We downloaded 26 ascending (satellite moving north looking east) SAR scenes and 32 descending (satellite moving south looking west) SAR scenes. During the snow-free period in 2018, only one ascending and descending track SAR scene was available. Each SAR scene has a pixel size of 14.1 m in azimuth and 2.3 m in range. Ascending scenes come from track 122 and descending scenes come from track 27. Corresponding POD Precise Orbit Ephemerides data were downloaded from the ESA website

(https://qc.sentinel1.eo.esa.int/aux_poeorb/).

Interferograms were processed in the JPL InSAR Scientific Computing Environment (ISCE) version 2 software (Rosen et al., 2012) with two looks in range and one look in azimuth, resulting in a ~5 m by ~14 m pixel size. Within a given summer (i.e., snow-free period), all possible combinations of interferograms were processed (Fig A1; Table A1). Late summer scenes

were combined with scenes from the following summers to create year-long pairs that bridge the winter when the ground is covered in snow. In total we created 45 ascending pairs and 63 descending pairs for a total of 108 interferograms. The minimum time between pairs was 6 days and the maximum time was 756, with a median of 48 days. We used a Shuttle Radar Topography Mission (SRTM) DEM with ~30 m pixel spacing to remove topographic signal from the phase and to geocode the interferograms, resulting in a ~30 m pixel spacing. To improve spatial resolution, selected one-year

interferogram pairs were reprocessed with a USGS 3DEP DEM with 10 m pixel spacing. Interferograms were unwrapped using the Statistical-Cost, Network-Flow Algorithm for Phase Unwrapping (SNAPHU) (Chen and Zebker, 2002). We applied a correlation threshold of 0.3 during unwrapping to remove low-quality data.





Many of the interferograms suffered from topography-correlated atmospheric effects that obscured signals of rock glacier
deformation. These atmospheric effects result from differences in pressure, temperature, and relative humidity in the lower
troposphere, causing radar to refract variably over the land surface, and producing a 2-way phase delay (Bekaert et al.,
2015). We applied a tropospheric phase delay correction based on pressure, temperature, and humidity predictions from the
ERA-I global weather model, which outputs data at a spatial resolution of 80 km on a 6 hour interval (Bekaert et al., 2015).
The correction was done using the TRAIN software package (Toolbox for Reducing Atmospheric InSAR Noise) from
Bekaert et al. (2015).

The resulting InSAR velocity maps were used along with Google Earth imagery, the USGS 10 m DEM, and the previous
Uinta rock glacier inventory (Munroe, 2018) to generate an active rock glacier inventory in QGIS 3.10. Rock glaciers
displaying a clear and relatively high LOS velocity signal consistent with the downslope direction were considered active
(Fig. 2). Boundaries of rock glaciers were delineated on the basis of morphology and InSAR-derived movement pattern.
Slope, aspect, and elevation of features in the rock glacier inventory were calculated in QGIS from the 10 m DEM. Rock
glaciers were classified as lobate or tongue-shaped (Barsch, 1996) based on morphology, and as "North Uintas" or "South
Uintas" based on their location relative to the east-west trending spine of the mountain range (Fig. 2). A non-parametric
Kruskall-Wallis test was used to establish significance of differences between groups.

Average annual velocities for rock glaciers were calculated in QGIS using velocity maps derived from ascending and
descending stacks of 1 year interferograms (Fig. 2). Average LOS velocity magnitudes were calculated by taking the mean
of the absolute value of velocity values over the surface of each rock glacier. We use the velocity magnitude to remove
negative LOS values that are caused by motion towards the satellite. We then define the characteristic rock glacier velocities
as the 75th percentile of the absolute value of velocity values within the mapped rock glacier body. This approach was used
to deemphasize noisy areas that could contain erroneous high or low velocities and to highlight the velocity of the active part
of each rock glacier (Bayer et al., 2018; Handwerger et al. 2019). For both mean and 75th percentile velocity, two values
were generated for each rock glacier, one derived from the ascending and another from the descending stack. The larger of
the ascending and descending values is used to represent rock glacier velocity in our data analysis.

Cumulative displacement time series were constructed using ascending and descending interferogram sets for three
representative rock glaciers selected on the basis of their clear deformation signal in the average velocity stack: Grayling
Lake, Whiterocks River, and Rockflour Lake (Fig. 3). Time series were calculated for these rock glaciers using the Small
Baseline Subset (SBAS) method (Berardino et al., 2002; Schmidt and Bürgmann, 2003). Interferograms with low overall
coherence were manually removed from the time series. Stable local reference points for each rock glacier were chosen using
Google Earth imagery to identify nearby ridge tops and bedrock outcrops.





Climatic data relevant to the rock glacier inventory were obtained by two sources: Parameter-elevation Regressions on Independent Slopes Model (PRISM, PRISM Climate Group, Oregon State University, http://prism.oregonstate.edu) and Snow Telemetry (SNOTEL, https://www.wcc.nrcs.usda.gov/snow/). PRISM data were used to establish the temperature and precipitation envelope of the rock glacier inventory. We downloaded 30 year normal (1981–2010) mean temperature and precipitation rasters with 800 m pixel spacing for each month with the PRISM package in R and used them to calculate mean annual air temperature (MAAT) and precipitation (MAP) for each rock glacier in the inventory. SNOTEL data (1980–2020) were used to constrain meteorological conditions during the intervals represented by the cumulative displacement time series. The Chepeta SNOTEL, at an elevation of 3200 m, is 10.0 km from the Whiterocks River rock glacier (3460 m) and 8.1 km from the Rockflour Lake rock glacier (3301 m) (Fig. 2). The Five Points Lake SNOTEL, at an elevation of 3335 m, is 11.8 km from the Grayling Lake rock glacier (3101 m).

## 3 Results

### 3.1 Rock glacier inventory

Our inventory contains 255 active rock glaciers totalling 27.6 km$^2$ that are found at an average elevation of 3290 ± 168 m (all ± corresponds to 1std, Fig. 4d). Rock glaciers average 10.8 ha in area; 6% are >30 ha, with a maximum of 60.2 ha (Fig. 4a). Their average slope is 22.0˚ ± 5.0˚ (Fig. 4g). Rock glaciers most frequently face north (20.3%), followed by northeast (16.1%), and east (12.2%) (Fig. 5a, b). For reference, steep slopes of the Uintas (>10˚) face generally north (18.7%) and south (19.6%) (Fig. 5c).

Morphologically, 79.2% of rock glaciers are lobate and 20.8% are tongue-shaped. Tongue-shaped rock glaciers are found at average elevations of 3357 ± 158 m, significantly higher than lobate rock glaciers, which are found at average elevations of 3273 ± 190 m (n = 255, p = 0.000, Fig. 4e). Lobate rock glaciers average 11.8 ha ± 11.2 ha in area, significantly larger than tongue-shaped rock glaciers, which are 7.30 ± 4.35 ha on average (n = 255, p = 0.039, Fig. 4b). All rock glaciers >20 ha in area are lobate rock glaciers. Morphology is not correlated with rock glacier slope or aspect.

A total of 59 rock glaciers (23.1% of the inventory) are located on the north side of the Uintas. These features are found at average elevation of 3343 ± 110 m, significantly higher than the 196 South Uintas rock glaciers, which are found at 3275 ± 180 m (n = 255, p = 0.0063, Fig. 4f). North Uintas rock glaciers are also significantly smaller than South Uintas rock glaciers, at 8.38 ± 8.39 ha on average, compared to 11.6 ± 10.7 ha (n = 255, p = 0.02, Fig. 4c) for South Uintas rock glaciers. Rock glaciers on the north side tend to face broadly northeast, with 50.9% facing north and 26.4% facing northeast, while rock glaciers on the south side do not appear to prefer a specific aspect. Slope angle is not significantly different for North and South Uintas rock glaciers (Fig. 4i). Tongue-shaped rock glaciers make up a larger proportion of North Uintas rock glaciers (28.8%) than the South Uintas rock glaciers (18.4%).





### 3.2 Rock glacier temperature and precipitation

Rock glaciers are found at an average MAAT of -0.15 ± 1.07 ˚C (Fig. 4j). 91.7% of rock glaciers are found at MAATs above -1.5 ˚C. Average MAAT at lobate rock glaciers is -0.06 ± 1.0 ˚C, significantly higher than tongue-shaped rock glaciers, which are found at an average MAAT of -0.50 ± 1.2 ˚C (n = 255, p = 0.002, Fig. 4k).

Rock glaciers receive an average of 75.2 ± 8.13 cm of precipitation per year, (Fig. 4m) of which ~29–40% is rain and ~60–71% is snow. North Uinta rock glaciers receive on average 80.4 ± 8.12 cm, significantly more than South Uinta rock glaciers, which receive 73.6 ± 7.71 cm (n = 255, p = 0.000, Fig. 4o).

### 3.3 Inactive rock glaciers

Of the rock glaciers identified by the previous inventory (Munroe, 2018), 155 are inactive based on our InSAR velocity maps. These inactive rock glaciers, which total 9.83 km$^2$, average 6.3 ± 6.6 ha in area, and are found at an average elevation of 3,281 ± 137 m (Fig. 6a, b). Their average MAAT is -0.031 ± 0.88 ˚C, and average MAP is 79.1 ± 8.95 cm (Fig. 6c, d). They are significantly smaller than active rock glaciers, and receive significantly more precipitation (area, n = 410, p = 0.000; precipitation, n = 410, p = 0.000, Fig. 6a, d). They also tend to experience higher MAATs (n = 410, p = 0.066, Fig. 6c). They face north most commonly, at 20.6% of the time, followed by south and west, both 15.5% of the time. They face north more commonly than active rock glaciers, which face north 13.3% of the time, and most commonly face northwest.

### 3.4 Rock glacier velocity

Rock glaciers generally have non-uniform spatial velocity patterns. On their rooting zone, rock glacier velocity tends to be highly heterogenous, varying over ~10 m length scales. On the main body of the rock glaciers, velocity is typically more homogenous, with one or more high-velocity patches surrounded by areas of a lower velocity. The characteristic LOS velocity from 2016–2019 was 2.52 ± 0.87 cm/yr (Fig. 7b). The fastest rock glacier velocities occurred during August each year, when several rock glaciers had maxima above 40 cm/yr. No metric of rock glacier velocity is significantly correlated with rock glacier area, elevation, slope, aspect, or morphology (Fig. 7a, Fig. A2).

Time series for the three representative rock glaciers show time-dependent deformation from 2016–2019 (Fig. 8a, b, c). Each of these rock glaciers had faster LOS velocities during snow-free observation periods than during snow-covered periods (n =30, p = 0.001, Fig. 8d). For example, the Grayling Lake rock glacier averaged 4.97 cm/yr during snow-free observation periods and 0.83 cm/yr during snow-covered periods.

In addition to seasonal changes in velocity, the Rockflour Lake rock glacier experienced a strong deceleration throughout 2018 (Fig. 8c). This rock glacier appears to have ceased motion the winter of 2017/18 and only displaced 0.11 cm during the





subsequent summer, fall, and winter of 2018. Yet during the same time period in 2016 and 2017, this rock glacier displaced 4.0 cm and 1.79 cm, respectively. This change in behavior corresponds temporally with a reduced spring melt caused by a small winter snowpack (Fig. 9).

## 4 Discussion

### 4.1 The rock glacier niche

The rock glacier inventory we created reveals the locations of active rock glaciers in the Uintas and illuminates the conditions necessary for rock glacier formation and sustained activity. Although the Uintas extend from elevations of ~2200 m to > 4100 m, 77.6% of rock glaciers are found in a narrow elevation band between 3100 and 3500 m. This zone has a MAAT from 0.5 to -2 °C, where talus production is high due to frost-cracking (Hales and Roering, 2007; Rempel et al.,

2016). Deposition of fresh talus onto the rock glacier source area helps sustain rock glacier motion (Barsch, 1996; Müller et al., 2016). The comparison between our active rock glacier inventory and the inactive rock glaciers identified by the previous inventory (Munroe, 2018) demonstrates the sensitivity of Uinta rock glaciers to temperature. Active rock glaciers are found at similar elevations to inactive rock glaciers, but active rock glaciers have a subzero median MAAT, whereas inactive rock glaciers have a median MAAT above zero (Fig. 6c). Rock glaciers can become inactive for myriad reasons, but almost all

are related to insufficient internal ice or insufficient talus supply (Barsch, 1996). Both talus production and stability of internal ice are related to temperature, as freeze-thaw cycles strongly control erosion and temperature is the primary control on permafrost stability (Hales and Roering, 2007; Hinzman, 1998). Depending to some extent on local conditions, permafrost begins to degrade at MAATs above 0 °C (Hoelzle and Haeberli, 1995), reducing ice content and rendering rock glaciers inactive.


Tongue-shaped rock glaciers are found at significantly higher elevations than lobate rock glaciers. Noting a similar trend in the Colorado Front Range, Janke (2007) concluded that high-elevation tongue-shaped rock glaciers form as ice glaciers are covered by debris. We cannot evaluate this theory with our data; however, in the Uintas, well-developed cirques are found mostly at high elevations (>3250 m), while steep valley walls continue down below the elevation where active rock glaciers

are observed (Munroe and Laabs, 2009). Tongue-shaped rock glaciers are mostly found in these high-elevation cirques, which, as a function of their geometry, may simply be more likely to produce tongue-shaped rock glaciers if debris is supplied from a narrow zone at the base of the cirque headwall (Degenhardt, 2009). In contrast, lobate rock glaciers are mostly found along valley walls, where debris is produced more consistently across a wider area (Degenhardt, 2009).

The geometry of the Uintas can explain other trends in rock glacier location. For instance, active rock glaciers in the North Uintas preferentially face northeast, while active rock glaciers in the South Uintas appear to have no strong preference for aspect. This north-facing trend in North Uintas rock glaciers is likely related to slope patterns. The North Uintas are steeper





than the South Uintas; as such, high elevation terrain is less extensive, and is found closer to the main east-west trending spine of the range. Thus, most rock glaciers in the North Uintas root on the east-west trending spine, and flow northward,

down-gradient. In the South Uintas, active rock glaciers face all directions in relatively equal proportions. Given that north-facing slopes are less common than south facing slopes on the south side of the range, South Uintas rock glaciers do face north more often than expected if rock glaciers were distributed randomly over steep mountain slopes. Rock glaciers may preferentially face north due to decreased sunlight exposure on north-facing slopes, which decreases local temperatures (Munroe, 2018).


The interaction between Uinta geometry and temperature further seems to control rock glacier morphology. In the North Uintas, 28.8% of rock glaciers are tongue-shaped, while in the South Uintas, 18.4% of rock glaciers are tongue shaped. In the gently sloping South Uintas, even valley walls far from the central spine of the mountains are at low enough temperatures to support rock glaciers, while in the steeper North Uintas, valley wall sites an equal distance from the crest

have MAATs well above 0 ˚C, and cannot support rock glaciers (Fig. 2b). These valley walls are associated with more lobate rock glaciers than tongue-shaped rock glaciers. The well-developed cirques associated with tongue-shaped rock glaciers, on the other hand, do not extend far from the spine of the Uintas. Thus, in the South Uintas, lobate rock glaciers are more common than tongue-shaped rock glaciers. In the steeper North Uintas, cirques and valley walls are in more equal proportion in the subzero MAAT zone, explaining the increased relative abundance of tongue-shaped rock glaciers. Since tongue-

shaped rock glaciers are smaller than lobate rock glaciers on average, this also helps to explain why North Uinta rock glaciers are significantly smaller than South Uinta rock glaciers.

## 4.2 Rock glacier velocity

Active rock glaciers in the Uintas deform at LOS rates between 0.88 and 5.26 cm/yr, with the average feature deforming 2.52 cm/yr. This range of mean velocities is lower than velocities reported for other rock glaciers in the western US and

around the world (Janke et al. 2005, Delaloye et al., 2010). We note, however, that rock glacier velocities are often calculated using different methods. Our velocity measurements correspond to average annual rock glacier velocity; thus, they average over seasonal changes such as the typical winter deceleration. Additionally, LOS velocity is an underestimate of true 3D velocity. Yet even after accounting for underestimates produced by our methods, it remains notable that rock glaciers in the Uintas appear to be moving nearly an order of magnitude slower than most other North American rock

glaciers (Janke et al. 2005).

Rock glaciers are sensitive to numerous variables that could differ between mountain ranges, possibly influencing rock glacier velocity; these include temperature and precipitation regime (Ikeda, et al., 2008; Eriksen et al., 2018) and talus characteristics and delivery rate (Arenson et al., 2002; Müller et al., 2016). However, despite the broad range of temperature

and precipitation conditions throughout the Uintas, there is no apparent relationship between velocity and local MAAT or





MAP. Furthermore, if talus characteristics have a strong influence on Uinta rock glacier velocities, we would expect adjacent rock glaciers which receive talus derived from the same bedrock units to have similar velocities. However, slow rock glacier velocities were observed throughout the Uintas, with little local grouping of similar velocities in a manner reflecting bedrock composition (Fig. 7a). Thus, the generally low velocities measured for Uinta rock glaciers are not likely due to consistent

differences in climate or talus delivery relative to other mountain ranges.  An alternative explanation for low rock glacier velocities in the Uintas is comparatively smaller quantities of internal ice. Most Uinta rock glaciers are found at MAATs close to 0 ˚C, based on 30 year temperature normals from 1980–2010. Thus, many rock glaciers may have experienced a negative mass balance of internal ice as temperatures warmed over the past decade.

Overall, Uinta rock glacier kinematics are complex and defy simple correlation with rock glacier area, elevation, slope, aspect, or morphology (Fig. A2). These factors likely interact to influence rock glacier rheology alongside additional variables including talus, snow, and meltwater delivery rate, clast size, underlying bedrock geometry, temperature, thickness, pore space, ice content, and shear zone geometry (Haeberli et al., 2006). Complex relationships between kinematics and external and internal factors make it especially challenging to predict rock glacier motion. This reality underscores the

benefit of using InSAR (or other remote sensing based kinematic data) to generate rock glacier inventories, allowing for quantifiable observation of rock glacier displacement.

Nonetheless, we encountered some limitations of InSAR that require consideration. First, the insensitivity of InSAR to movement along flight direction (i.e., perpendicular to the LOS direction) made it difficult to determine whether some north

or south facing rock glaciers were actively deforming. This may have produced a bias in our inventory toward rock glaciers flowing broadly eastward or westward. In the prior Uinta Rock glacier inventory based on morphological data (Munroe, 2018), 20.6% of rock glaciers face north, while 15.5% of rock glaciers faced south. In our active rock glacier inventory, 13.3% of rock glaciers faced north, and 8.62% of rock glaciers faced south. While south-facing rock glaciers may be more likely to become inactive due to increased sunlight exposure, elevated local temperatures, and consequent melting of internal

ice, it is reasonable to expect similar proportions of north-facing rock glaciers in both inventories. This was not observed, which suggests a bias against north and south facing rock glaciers in our inventory. Second, the LOS measurements provided by InSAR underestimate the true 3D velocity of rock glaciers. For rock glaciers with a larger relative component of northward or southward motion, this underestimate is more extreme. In our inventory, North Uintas rock glaciers, which preferentially face north, have significantly lower velocities than South Uintas rock glaciers, which face more equally in all

directions.  Finally, InSAR unwrapping errors, which are common in areas with high deformation rates and gradients, also likely introduced inaccuracies in our velocity estimates. Several of the fastest rock glaciers move at rates up to 40 cm/yr in 12 day interferograms, but have velocities < 4 cm/yr in the 1 year pairs used to calculate average velocity. While these discrepancies could be the result of particularly strong seasonal changes in velocity, in longer baseline interferograms, these rock glaciers sometimes had sharp velocity discontinuities consistent with phase jumps caused by unwrapping errors.





However, due to the slow velocities of most rock glaciers in the inventory, any unwrapping errors are unlikely to have a large impact on our velocity estimates.

### 4.3 Time-dependent rock glacier deformation

Time series constructed from overlapping sets of interferograms provide insight into how rock glacier velocity varies over three superimposed time scales: 1) seasonal, 2) yearly, and 3) multi-year. Over the four years that we measured
displacement, the Grayling Lake, Whiterocks River, and Rockflour Lake rock glaciers displayed continuous motion with no multi-year trend (Fig. 8). It's likely that this observation period was too small to capture possible long-term trends in rock glacier motion, as have been well-documented in the Alps (Delaloye et al., 2008; Kääb et al., 2007; Kaufmann and Ladstädter, 2007; Roer et al., 2005; Vonder Muehll et al., 2007).

On the other hand, Uinta rock glaciers do exhibit a significant seasonal velocity pattern. The three representative rock glaciers moved at an average rate of 4.42 cm/yr during the snow-free period from July to October; the rest of the year, they moved at a rate of 0.86 cm/yr. This seasonal rhythm illustrates that Uinta rock glaciers are responsive to the short-term changes in conditions, such as snowmelt.

Examining velocities of individual rock glaciers over time clarifies how they respond to changes in meltwater delivery. For example, the Rockflour Lake rock glacier moved only 0.11 cm in the LOS direction from the winter of 2017/18 to the spring of 2019. While near-zero velocities occurred sporadically for multiple rock glaciers during the snow-covered winter months, the absence of any meaningful acceleration of this rock glacier during the following spring, summer, fall, and winter is unique in our dataset. Spring and summer acceleration in rock glaciers is understood to depend on meltwater and
precipitation infiltration to the rock glacier shear zone, increasing pore pressure and decreasing the overall material strength (Wirz et al., 2016; Kenner et al., 2017; Cicoira et al., 2019; Fey and Krainer, 2020). During the winter of 2017/18, snow-water equivalent (SWE) at the Chepeta SNOTEL station, close to the Rockflour Lake rock glacier, was low, and air temperatures were similar to other winters. This correspondence suggests that spring acceleration of Uinta rock glaciers is controlled by snowmelt and that inadequate meltwater infiltration during the spring and summer may cause a rock glacier to
effectively "skip" the summer portion of its seasonal cycle. Reduced spring infiltration of liquid water to the shear zone of the Rockflour Lake rock glacier could explain how even as air and ground temperature rose, this feature did not move a measurable distance during the summer and fall of 2018.

Both the Grayling Lake and Whiterocks River rock glaciers accelerated normally during the warmer months of 2018. The
Grayling Lake rock glacier is close to the Five Points Lake SNOTEL station, which received around 15 cm more SWE than the Chepeta SNOTEL station during the winter of 2017/18 and had a more robust snowmelt event in the spring. Thus, this rock glacier likely received the requisite amount of meltwater infiltration for normal acceleration. The Whiterocks River rock





glacier is closest to the Chepeta SNOTEL station, but is 260 m higher in elevation. Snowpack at the Whiterocks River rock glacier may have been much larger than at the Chepeta station, which could explain its normal spring acceleration. Alternatively, the Whiterocks River rock glacier could require less water to accelerate in the spring.

Collectively, these results illustrate that Uinta rock glaciers are sensitive to changes in climatic conditions, though individual rock glaciers did not respond uniformly to reduced snowmelt in 2018. Our data are not sufficient to determine whether dissimilar responses were caused by local variations in climate that weren't reflected in our regional weather data, or by individual rock glaciers responding differently to similar climate events. Time series analysis of additional rock glaciers with high-resolution weather data would aid our understanding of Uinta rock glaciers' response to climate events.

### 4.4 Rock glaciers and climate change

Our results suggest that Uinta rock glaciers are vulnerable to loss of internal ice due to climate change. Currently, 91.7% of these features are found at MAATs above -1.5 °C. If warming continues as predicted, all of these rock glaciers will have MAATs at or above 0 °C in the next ten years (IPCC, 2018). Because the PRISM temperature data used here are effectively 25 years old, they underestimate current MAATs. PRISM 30 year normals provide a MAAT of 0.11 °C at the Chepeta SNOTEL and 0.52 °C at the Five Points Lake SNOTEL, but measured MAAT over the past five years at these stations has been warmer, at 1.35 °C and 0.97 °C respectively. It is likely that almost all Uinta rock glaciers are experiencing significant ice loss in response to rising temperatures. The presence of 155 inactive rock glaciers supports this claim.

Further support for this conclusion comes from the slow velocities of Uinta rock glaciers. In the European Alps and Norway, warming temperatures have been associated with rock glacier acceleration and destabilization, as newly established channels in permafrost direct meltwater and rain to the shear zone (Delaloye et al., 2008; Kääb et al., 2007; Kaufmann and Ladstädter, 2007; Roer et al., 2005; Vonder Muehll et al., 2007, Eriksen et al., 2018). However, in these locales, precipitation is generally increasing, and rainfall is replacing snowfall. In contrast, at the Five Points Lake and Chepeta SNOTEL station in the Uintas, MAP has decreased over the past 40 years, and rainfall has decreased at a faster rate than snowfall (Fig. 10g, h). In the Uintas, and much of the western United States, climate change is characterized by increasing aridity (Gutzler and Robbins, 2011). This aridity could explain the slow velocities of Uinta rock glaciers, as 1) reduced winter snowpack means that spring accelerations are less extreme (Fig. 10a, b), and 2) reduced rainfall results in less heat pumped into the rock glacier interior, and less water in the shear zone, further reducing rock glacier velocity. Thus, rock glacier degradation in the Uintas is unlikely to manifest as destabilization and acceleration. Instead, degrading permafrost may manifest as progressively slower rock glacier velocities and eventual inactivation, as decreasing ice content, increasing frictional interactions from liberated debris, and decreased shear stress as a result of reduced rock glacier thickness serve to slow the rock glacier. These processes can explain the slow velocities of Uinta rock glaciers, and may indicate that these features are not as vulnerable to rapid ice loss as those in parts of the world with increasing precipitation and rain. Uinta rock glaciers





may, therefore, serve as a water resource for a longer period of time relative to their ice content and are unlikely to undergo hazardous destabilization.

### 4.5 Implications for hydrology

Following calculations in Munroe (2018), which assume a rock glacier and talus thickness of 10 m, an average porosity of
30%, and 25% ice saturation of pore space, active rock glaciers mapped in this study contain about $2.07 \times 10^7$ m$^3$ of water, equal to about 1% of annual runoff from the Uintas (Jeppson, 1968). Other work has used less conservative numbers for ice content of rock glaciers. For example, Janke et al., (2017) estimated ice content of active rock glaciers in the semiarid Aconcagua River Basin of Chile to be between 25–44.9%, and Jones et al. (2018b) estimated ice content of active rock glaciers in the Nepalese Himalaya to be between 40% and 60%. Using these estimates, active rock glaciers in the Uintas
could store $6.9 \times 10^7$–$1.24 \times 10^8$ m$^3$ of water, up to 9.21% of the annual runoff from these mountains. These calculations illustrate that rock glaciers in the Uintas could store and seasonally release non-trivial quantities of water. However, if Uinta rock glaciers have experienced significant loss of internal ice due to climate change, water storage will be on the lower end of this spectrum.

While the magnitude of this release may be small compared to the total amount of runoff from the Uintas, it likely makes up a much larger portion of runoff in the high-elevation areas. The timing of meltwater release is also likely to be important. Uinta rock glaciers melt at the fastest rate in the summer, when temperatures are highest, and rainfall delivers liquid water to interior of the rock glacier. In the late summer, when precipitation in the Uintas is scarce, rock glacier meltwater probably contributes a larger proportion of total streamflow (MacDonald and Tingstad, 2007). As noted by Munroe (2018), if rock
glaciers provide a significant input to late summer base flow, climate change will likely produce a short-term increase in water availability, as rock glaciers melt at increased rates, followed by a long term decrease in water availability (Clow et al. 2003).

### 5 Conclusions

In this study an inventory of 255 active rock glaciers in the Uinta Mountains, Utah was constructed using InSAR from the
Copernicus Sentinel-1 satellites, Google Earth imagery, and topographic data. Rock glaciers in the Uinta Mountains have an average area of 10.8 ha, commonly face north or northeast, and are found within a narrow elevation band ranging from 3100 and 3500 (average of 3290 m), corresponding to a MAAT of 0.5 to -2°C (average of -0.15°C). Rock glacier location is controlled by a combination of mountain geometry and air temperature. Rock glacier velocities are between 0.88 and 5.26 cm/yr, with an average of 2.52 cm/yr, and are not correlated with variables including elevation, area, aspect, slope, or
morphology. Time series analysis of three rock glaciers revealed a seasonal rhythm in velocities, with an average of 4.42 cm/yr during the snow-free late summer, and 0.86 cm/yr during the rest of the year. The Rockflour Lake rock glacier did not accelerate during 2018 because of minimal spring snowmelt following a winter with abnormally low snowfall. We speculate

that generally slow rock glacier velocities throughout the Uintas may be related to reduced ice content and increasing aridity caused by ongoing climate change. We estimate that rock glaciers in the Uintas store a volume of water equivalent to 1-10%

of annual runoff from these mountains. During the late summer, when precipitation over the Uintas is comparatively low, rock glacier melting may provide an important addition to stream base flow, particularly at high elevations. As climate change continues, the rock glacier contribution to late summer runoff is expected to increase, as rock glaciers melt, before decreasing in the long term, as Uinta rock glaciers become relict.

**Data Availability**

The research data used in this study are freely available online. SAR scenes used to generate interferograms are available on the Alaska Satellite Foundation Vertex website (https://vertex.daac.asf.alaska.edu/). Corresponding POD Precise Orbit Ephemerides data can be found on the ESA website (https://qc.sentinel1.eo.esa.int/aux_poeorb/). The 30 m SRTM DEM is available from the USGS Earth Explorer website (https://earthexplorer.usgs.gov/), and the 10 m USGS 3DEP DEM is

available from the USGS National Map website (https://viewer.nationalmap.gov/basic/). PRISM data can be downloaded from the PRISM website (https://prism.oregonstate.edu/). SNOTEL data are available on the Natural Resources Conservation Service website (https://www.nrcs.usda.gov/wps/portal/wcc/home/).

**Author Contribution**

J.M. devised the idea for the project. A.H. and G.B. designed the methodology. G.B. carried out the methodology with guidance from A.H. G.B. analysed the data. G.B. made the figures with advice from J.M. and A.H. G.B. wrote the manuscript with significant input and editing from J.M. and A.H.

**Conflicts of Interest**

The authors declare that they have no conflict of interest.

**Acknowledgements**

Funding was provided by National Science Foundation award EAR-1935200 to A. Handwerger and J. Munroe. Part of this research was carried out at the Jet Propulsion Laboratory, California Institute of Technology, under a contract with the

National Aeronautics and Space Administration (80NM0018D0004). We'd like to thank the scientists at the Jet Propulsion Laboratory for generously providing their insight during our research visit. Funding to support that visit was awarded to George Brencher by Middlebury College's Senior Research Project Supplement.



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





**Figure 1.** Figure showing general internal structure and deformation of rock glaciers. (a) Oblique view of a Uinta Mountain rock glacier from © Google Earth. (b) Schematic stratigraphic cross section of a rock glacier in the Uinta Mountains. Schematic diagram is based on borehole data from Arenson et al. (2002). (c) Plot showing how total displacement is related to shearing and plastic deformation in rock glaciers. Deformation primarily occurs in shear zones at the base of the rock glacier. (d) Schematic showing rock glacier deformation ((c) and (d) after Kenner et al. 2017).




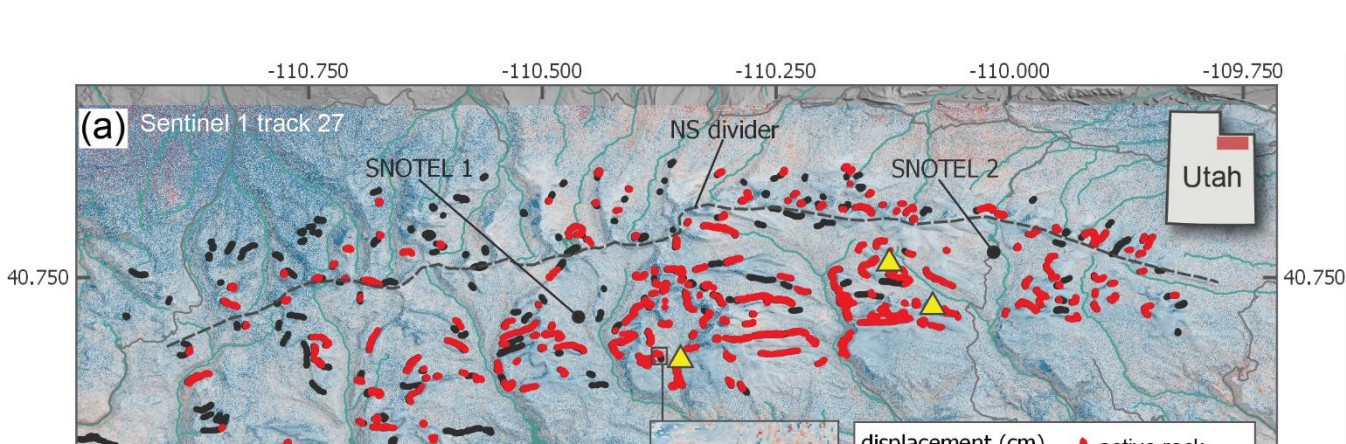

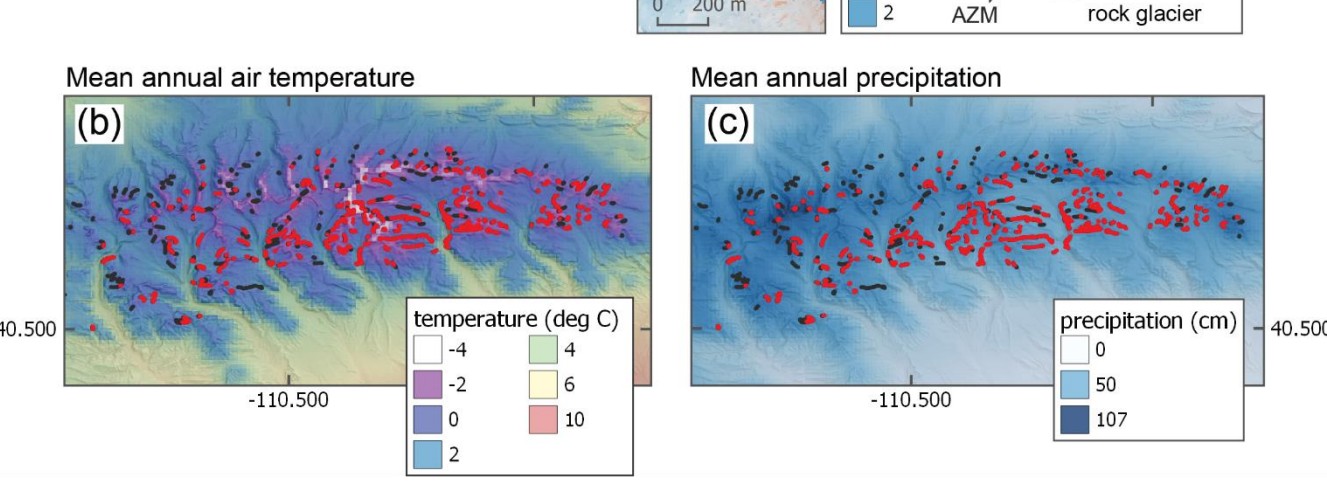

**Figure 2.** Uinta Mountains study site. (a) Hillshade map of the Uinta Mountains overlaid with average InSAR velocity from descending track 27. Red polygons represent active rock glaciers identified in this study. Black polygons represent inactive rock glaciers identified in the previous inventory (Munroe, 2018). Rock glacier size is exaggerated for plotting. SNOTEL 1 is the Five Points Lake station. SNOTEL 2 is the Chepeta station. The black dashed line separates the "North Uintas" from the "South Uintas". Yellow triangles represent the three rock glaciers used in time series analysis, which are, from left to right, the Grayling Lake, Whiterocks River, and Rockflour Lake rock glaciers. Data from USGS, NRCS, and Natural Earth. (b) Mean annual air temperature of the Uinta Mountains. (c) Mean annual precipitation in the Uinta Mountains. In (b) and (c), data are derived from PRISM 30 year normals (1981-2010).







**Figure 3.** Individual rock glaciers investigated with time series. (top) Oblique 3D view of rock glaciers in © Google Earth, annotated and clipped in Adobe Illustrator. (bottom) Average velocity stack draped over a hillshade in QGIS. Rock glaciers were added to our active inventory if they displayed clear deformation signal, like those above.




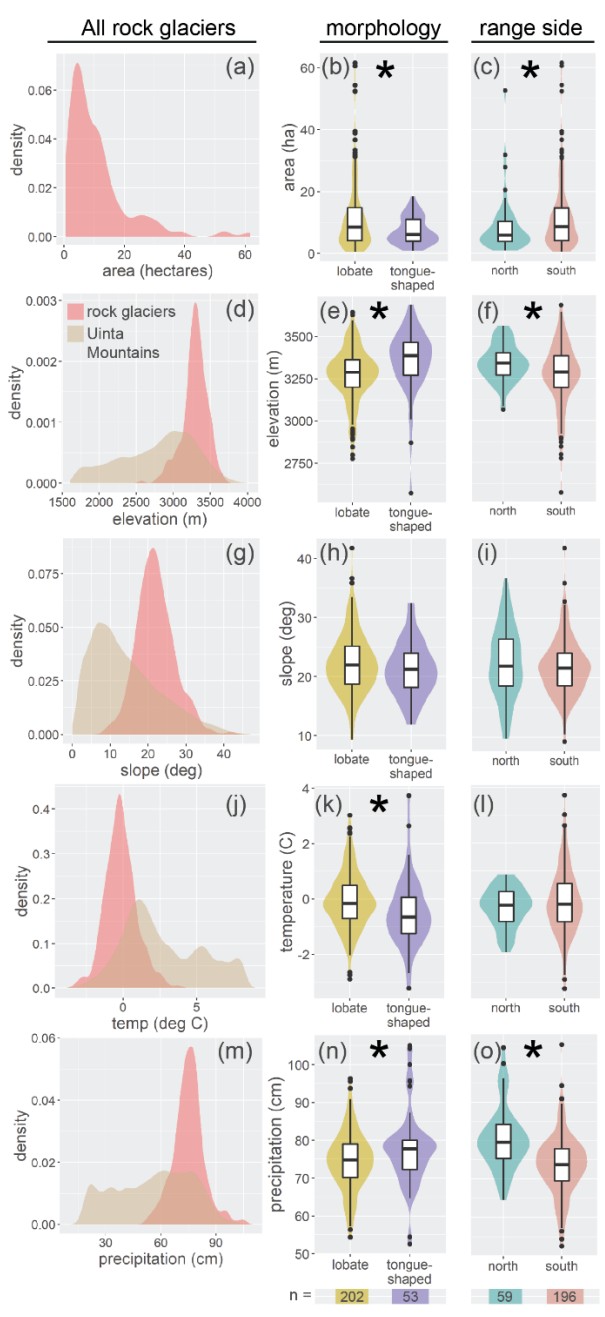

**Figure 4.** Characteristics of active rock glaciers in the Uinta Mountains. Slope, area, and elevation were calculated using the rock glacier inventory polygons and a DEM in QGIS. Temperature and precipitation were calculated using the rock glacier inventory polygons and PRISM data in QGIS. (a, d, g, j, m) Data for all rock glaciers and the Uinta Mountains shown as kernel density plots. (b, c, e, f, h, i, k, l, n, o) Colored violin plots on the right show rock glacier characteristics based on morphology and location within the Uintas. Asterisks denote statistically significant differences. n = 255.



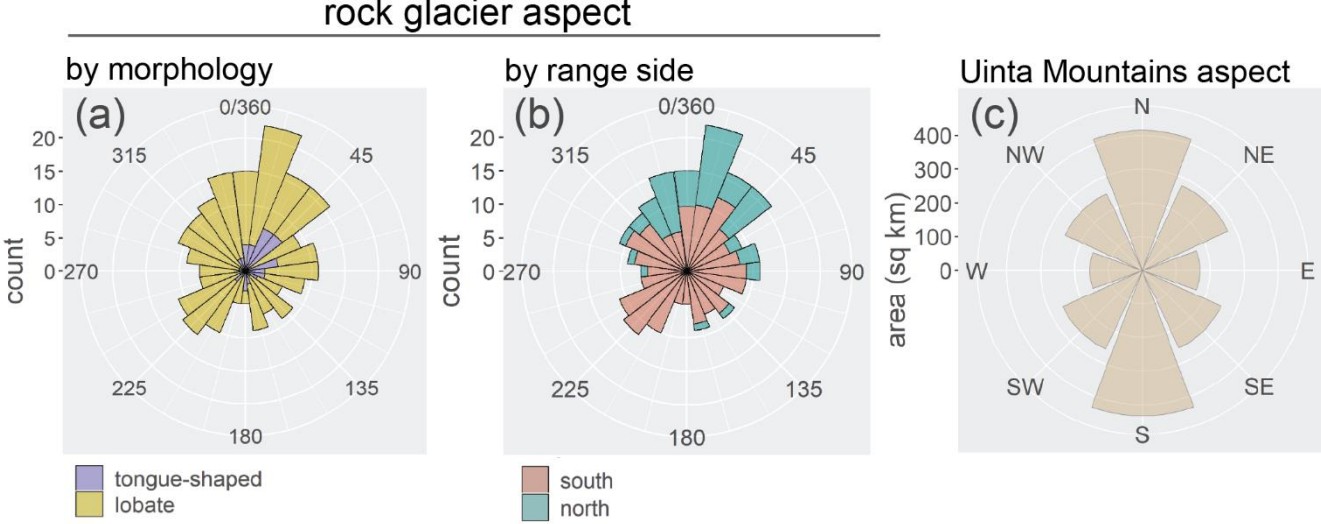

**Figure 5.** Polar plots showing aspect of active rock glaciers from the Uinta Mountains inventory and aspect of the steep slopes (>10°) of the Uinta Mountains. (a) Aspect is classified by rock glacier morphology. (b) Aspect is classified by which side of the Uinta Mountains the rock glacier is on. Data are derived from rock glacier inventory polygons and a USGS DEM.


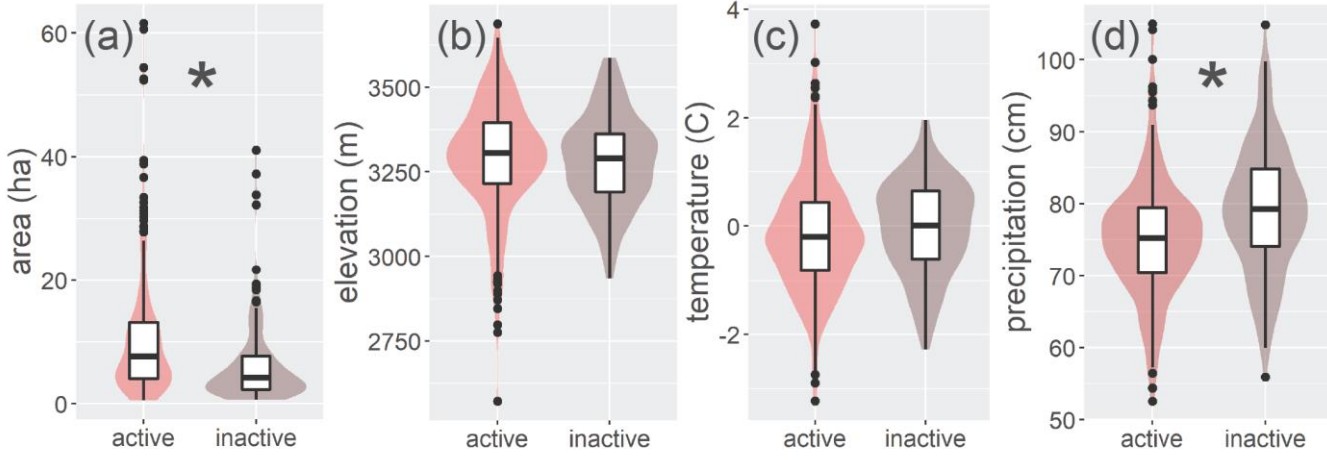

**Figure 6.** Violin plots comparing active rock glaciers with inactive rock glaciers from the previous Uinta rock glaciers inventory. Asterisks denote statistically significant differences. n = 410






**Figure 7.** Velocity of active Uinta rock glaciers from 2016-2019. (a) Map showing rock glacier area and velocity. Each rock glacier is represented by a circle. Size of circle represents relative rock glacier area and color represents absolute value of rock glacier 75th percentile line-of-sight (LOS) velocity. Note fewer, smaller, slower rock glaciers in the North Uintas. (b) Violin plots show absolute value of mean and 75th percentile LOS velocities for all rock glaciers. (c) Violin plots show absolute value of mean and 75th percentile LOS velocities for rock glaciers classified by morphology. (d) Violin plots show absolute value of mean and 75th percentile LOS velocities for rock glaciers classified by which side of the Uinta Mountains they are on. Asterisks denote significant differences. n = 255.






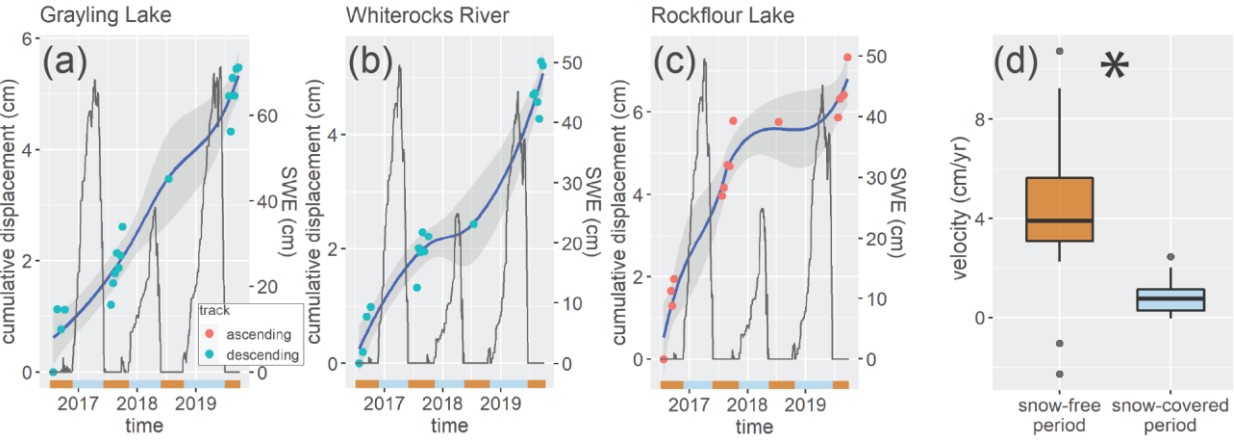

**Figure 8.** Velocity time series for three representative rock glaciers. (a, b, c) Time-dependent displacement of three rock glaciers from 2016-2019 and snow-water equivalent (SWE) from nearby weather stations. Blue and orange lines along the x-axes denotes snow-free (orange) and snow-covered (blue) periods. Ascending and descending LOS are scaled so they are both positive and increasing. Time series displacement values were calculated using an average of 9 pixels (8,100 m$^2$ total area) in the rock glacier. (d) LOS velocities of three rock glaciers during the snow-free period in late summer compared to the snow-covered period from October to July. n = 30, p = 0.001.


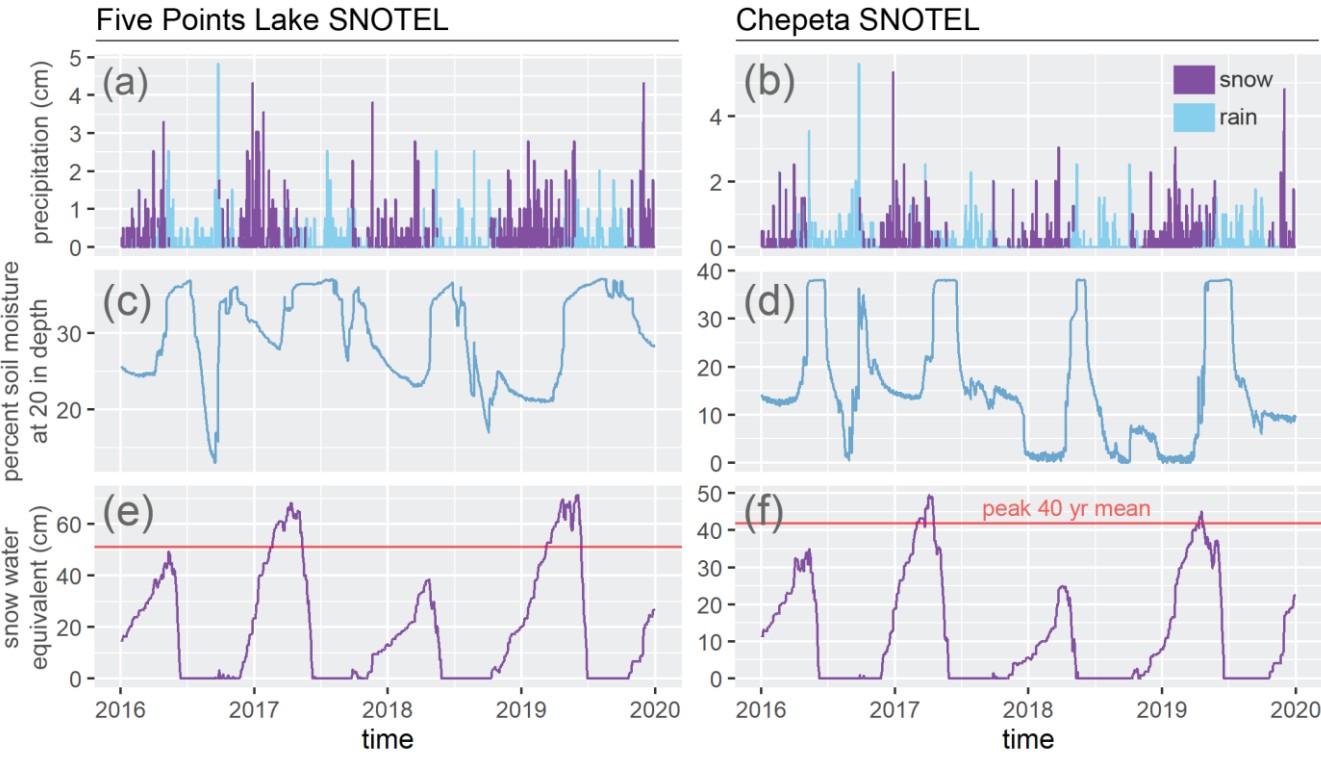

**Figure 9.** Precipitation, soil moisture, soil temperature, and snow water equivalent at the Five Points Lake and Chepeta SNOTEL stations from 2016–2019. (a, b) Precipitation events. Purple lines represent snow and blue lines represent rain. Precipitation type was inferred from average daily temperature at the SNOTEL station. (c, d) Percent soil moisture at 20 in depth. (e, f) Snow water equivalent (SWE). Red lines represent average peak SWE over the past 40 years.








**Figure 10.** Climate over the past 40 years at the Five Points Lake SNOTEL (left) and Chepeta (right) SNOTEL stations. (a, b) Snow water equivalent (SWE). The light blue line is daily SWE. The dark blue points are mean annual SWE. (c, d) Number of days with no snow cover. Blue points represent number of days per year where SWE is 0. (e, f) Temperature over time. Blue to red regions represent daily temperature. Grey points are mean annual air temperature. (g, h). Mean annual precipitation over time. Blue points represent mean annual precipitation. For all, red lines are linear trendlines, and gray regions around lines are 95% confidence intervals. Gray boxes represent the study period.



**Appendix A**

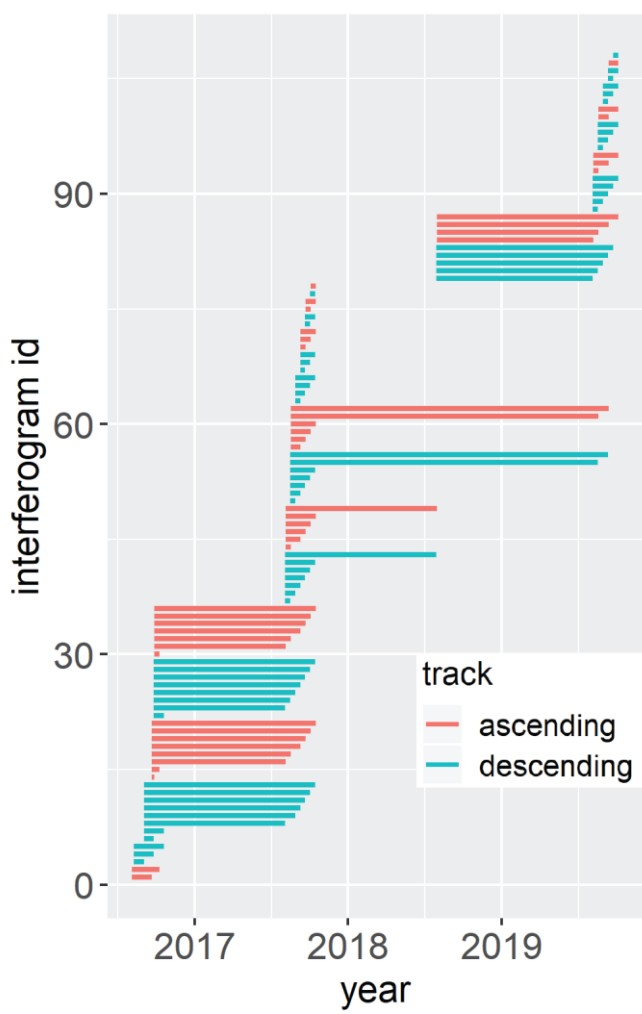

**Figure A1.** Temporal distribution of interferograms. Interferograms were generated using Sentinel 1 IW SLC scenes. Interferogram pairs were chosen based on snow-free periods. Red and blue lines represent ascending and descending interferograms, respectively.

**Table A1.** Track, date, and time span of all interferograms generated.

| number | track | primary acquisition date (yyymmdd) | secondary acquisition date (yyyymmdd) | timespan (days) |
|---|---|---|---|---|
| 1 | T122 ascending | 20160804 | 20160921 | 48 |
| 2 | T122 ascending | 20160921 | 20160927 | 6 |
| 3 | T122 ascending | 20160804 | 20161009 | 66 |





| 4 | T122 ascending | 20160921 | 20161009 | 18 |
|---|---|---|---|---|
| 5 | T122 ascending | 20160927 | 20161009 | 12 |
| 6 | T122 ascending | 20160921 | 20170805 | 318 |
| 7 | T122 ascending | 20160927 | 20170805 | 312 |
| 8 | T122 ascending | 20160921 | 20170817 | 330 |
| 9 | T122 ascending | 20160927 | 20170817 | 324 |
| 10 | T122 ascending | 20170805 | 20170817 | 12 |
| 11 | T122 ascending | 20160921 | 20170910 | 354 |
| 12 | T122 ascending | 20160927 | 20170910 | 348 |
| 13 | T122 ascending | 20170805 | 20170910 | 36 |
| 14 | T122 ascending | 20170817 | 20170910 | 24 |
| 15 | T122 ascending | 20160921 | 20170922 | 366 |
| 16 | T122 ascending | 20160927 | 20170922 | 360 |
| 17 | T122 ascending | 20170805 | 20170922 | 48 |
| 18 | T122 ascending | 20170817 | 20170922 | 36 |
| 19 | T122 ascending | 20170910 | 20170922 | 12 |
| 20 | T122 ascending | 20160921 | 20171004 | 378 |
| 21 | T122 ascending | 20160927 | 20171004 | 372 |
| 22 | T122 ascending | 20170805 | 20171004 | 60 |
| 23 | T122 ascending | 20170817 | 20171004 | 48 |
| 24 | T122 ascending | 20170910 | 20171004 | 24 |
| 25 | T122 ascending | 20170922 | 20171004 | 12 |
| 26 | T122 ascending | 20160921 | 20171016 | 390 |
| 27 | T122 ascending | 20160927 | 20171016 | 384 |
| 28 | T122 ascending | 20170805 | 20171016 | 72 |
| 29 | T122 ascending | 20170817 | 20171016 | 60 |
| 30 | T122 ascending | 20170910 | 20171016 | 36 |
| 31 | T122 ascending | 20170922 | 20171016 | 24 |
| 32 | T122 ascending | 20171004 | 20171016 | 12 |
| 33 | T122 ascending | 20170805 | 20180731 | 360 |
| 34 | T122 ascending | 20180731 | 20190807 | 372 |
| 35 | T122 ascending | 20170817 | 20190819 | 732 |
| 36 | T122 ascending | 20180731 | 20190819 | 384 |
| 37 | T122 ascending | 20190807 | 20190819 | 12 |
| 38 | T122 ascending | 20170817 | 20190912 | 756 |
| 39 | T122 ascending | 20180731 | 20190912 | 408 |
| 40 | T122 ascending | 20190807 | 20190912 | 36 |
| 41 | T122 ascending | 20190819 | 20190912 | 24 |
| 42 | T122 ascending | 20180731 | 20191006 | 432 |
| 43 | T122 ascending | 20190807 | 20191006 | 60 |





| 44 | T122 ascending | 20190819 | 20191006 | 48 |
| 45 | T122 ascending | 20190912 | 20191006 | 24 |
| 46 | T27 descending | 20160809 | 20160902 | 24 |
| 47 | T27 descending | 20160809 | 20160926 | 48 |
| 48 | T27 descending | 20160902 | 20160926 | 24 |
| 49 | T27 descending | 20160809 | 20161020 | 72 |
| 50 | T27 descending | 20160902 | 20161020 | 48 |
| 51 | T27 descending | 20160926 | 20161020 | 24 |
| 52 | T27 descending | 20160902 | 20170804 | 336 |
| 53 | T27 descending | 20160926 | 20170804 | 312 |
| 54 | T27 descending | 20160926 | 20170816 | 324 |
| 55 | T27 descending | 20170804 | 20170816 | 12 |
| 56 | T27 descending | 20160902 | 20170828 | 360 |
| 57 | T27 descending | 20160926 | 20170828 | 336 |
| 58 | T27 descending | 20170804 | 20170828 | 24 |
| 59 | T27 descending | 20170816 | 20170828 | 12 |
| 60 | T27 descending | 20160902 | 20170909 | 372 |
| 61 | T27 descending | 20160926 | 20170909 | 348 |
| 62 | T27 descending | 20170804 | 20170909 | 36 |
| 63 | T27 descending | 20170816 | 20170909 | 24 |
| 64 | T27 descending | 20170828 | 20170909 | 12 |
| 65 | T27 descending | 20160902 | 20170921 | 384 |
| 66 | T27 descending | 20160926 | 20170921 | 360 |
| 67 | T27 descending | 20170804 | 20170921 | 48 |
| 68 | T27 descending | 20170816 | 20170921 | 36 |
| 69 | T27 descending | 20170828 | 20170921 | 24 |
| 70 | T27 descending | 20170909 | 20170921 | 12 |
| 71 | T27 descending | 20160902 | 20171003 | 396 |
| 72 | T27 descending | 20160926 | 20171003 | 372 |
| 73 | T27 descending | 20170804 | 20171003 | 60 |
| 74 | T27 descending | 20170816 | 20171003 | 48 |
| 75 | T27 descending | 20170828 | 20171003 | 36 |
| 76 | T27 descending | 20170909 | 20171003 | 24 |
| 77 | T27 descending | 20170921 | 20171003 | 12 |
| 78 | T27 descending | 20160902 | 20171015 | 408 |
| 79 | T27 descending | 20160926 | 20171015 | 384 |
| 80 | T27 descending | 20170804 | 20171015 | 72 |
| 81 | T27 descending | 20170816 | 20171015 | 60 |
| 82 | T27 descending | 20170828 | 20171015 | 48 |
| 83 | T27 descending | 20170909 | 20171015 | 36 |





| 84 | T27 descending | 20170921 | 20171015 | 24 |
| 85 | T27 descending | 20171003 | 20171015 | 12 |
| 86 | T27 descending | 20170804 | 20180730 | 360 |
| 87 | T27 descending | 20180730 | 20190806 | 372 |
| 88 | T27 descending | 20170816 | 20190818 | 732 |
| 89 | T27 descending | 20180730 | 20190818 | 384 |
| 90 | T27 descending | 20190806 | 20190818 | 12 |
| 91 | T27 descending | 20180730 | 20190830 | 396 |
| 92 | T27 descending | 20190806 | 20190830 | 24 |
| 93 | T27 descending | 20190818 | 20190830 | 12 |
| 94 | T27 descending | 20170816 | 20190911 | 756 |
| 95 | T27 descending | 20180730 | 20190911 | 408 |
| 96 | T27 descending | 20190806 | 20190911 | 36 |
| 97 | T27 descending | 20190818 | 20190911 | 24 |
| 98 | T27 descending | 20190830 | 20190911 | 12 |
| 99 | T27 descending | 20180730 | 20190923 | 420 |
| 100 | T27 descending | 20190806 | 20190923 | 48 |
| 101 | T27 descending | 20190818 | 20190923 | 36 |
| 102 | T27 descending | 20190830 | 20190923 | 24 |
| 103 | T27 descending | 20190911 | 20190923 | 12 |
| 104 | T27 descending | 20190806 | 20191005 | 60 |
| 105 | T27 descending | 20190818 | 20191005 | 48 |
| 106 | T27 descending | 20190830 | 20191005 | 36 |
| 107 | T27 descending | 20190911 | 20191005 | 24 |
| 108 | T27 descending | 20190923 | 20191005 | 12 |






**Figure A2.** Rock glacier velocity plotted against physical variables and climate envelope. In all plots, velocity is the absolute value of 75th percentile LOS velocity, compared to (a) rock glacier area, (b), rock glacier elevation, (c) rock glacier slope, (d) rock glacier mean annual precipitation, (e) rock glacier mean annual air temperature, and (f) rock glacier aspect. There is no direct relationship between rock glacier velocity and the above listed rock glacier characteristics. Temperature and precipitation data are from PRISM. n = 255.