# Peer review of "InSAR-based characterization of rock glacier movement in the Uinta Mountains, Utah, USA"

_The Cryosphere, 2020_

## Referee Comment (RC1) · Anonymous Referee #1 · 12 Jan 2021

Overall Comments

The authors used InSAR technique to map and characterize rock glacier movement in a region where previous knowledge of rock glaciers is limited. It produces a new dataset that sheds light on the kinematic behavior of those permafrost landforms and provides interesting insights as to how the rock glaciers respond to the climatic conditions and their potential local hydrological importance in the future. Hopefully, this paper will be published and help generate more interest in studying rock glaciers in North America.

It is a well-written paper in general. However, I would raise a few issues mostly regarding the necessary details of the InSAR method adopted in this work. Accuracy in

terminology and clarity in the argument can be further improved in a few places. Please see my comments below.

Detailed comments

1. Line 21, the definition of rock glaciers here is inaccurate because they are not entirely "perennially frozen bodies", the upper part of which is seasonally frozen ground or the so-called active layer.

2. Line 32–34, it might be inappropriate to draw an analogy between rock glaciers and ice glaciers here, because some of the enumerated drivers (e.g., liquid water, pore water pressure) influence the motion of the two types of landforms in ways that can hardly be regarded as similar.

3. Line 46, shear horizon is NOT "at the base of the rock glacier". Borehole investigations have revealed that sediments exist below the shear horizon, though the motion of which is negligible. The authors may refer to the two papers cited in the caption of Figure 1(i.e., Arenson et al., 2002; Kenner et al., 2017) and modify Figure 1b and 1d accordingly.

4. Line 69–70, what are the "significant patterns"?

5. Line 124–125, why do the authors use the 10-m resolution DEM for selected one-year pairs only, instead of applying it to all data?

6. Line 138–139, the description "LOS velocity signal consistent with the downslope direction" is confusing, because a LOS signal is obviously always in the LOS direction, which is from the ground to the satellite, and thus cannot be consistent with downslope direction.

7. Line 147, which one-year interferograms do the authors use for calculating annual velocities? Here the authors mention both ascending and descending stacks of interferograms, however, Figure 2 only shows results derived from one descending track.

[Figure]

8. Line 149, why do the authors remove negative LOS values? The motion towards the satellite is possible and Figure 2a does include negative values.

9. In Figure 2a, Line 715–716, the authors should specify the time span they used to calculate the average velocity, instead of just providing satellite orbital information. Also, the legend shows the unit of the velocity map in distance unit (cm) which may confuse the readers. Is Figure 2a a displacement map or a velocity map?

10. In Figure 2a, Line 715, the legend shows the unit of the velocity map is in centimeters which may confuse the readers. And the period of the observation should be specified.

11. Line 163–172, this part is not under the topic of "InSAR analysis". The authors may consider reorganizing the structure of this section. Please also refer to the first technical correction below.

12. Line 236–239 and Figure 2, Line 716, the previous inventory (Munroe, 2018) didn't classify the mapped rock glaciers based on their activities. How do the authors identify the inactive rock glaciers from the previously published dataset? If the inactive rock glaciers are landforms that do not show displacement in the interferograms, is it possible that some of those landforms are actually active, but their activity is not detected by InSAR, due to limitations of the technique, such as decorrelation, shadow, overlay, or the flow direction of landform is insensitive to the LOS direction?

13. Line 279–280, the references here do not fully fit. Delaloye et al. (2010) focus on the Swiss Alps which is a regional study and cannot represent rock glaciers "around the world".

14. Line 283–285, Janke et al. (2005) reported average velocities of 7.3, 6.3, and 9.5 cm/yr for three rock glaciers in the Front Range, which are not notably faster than the LOS rates between 0.88 and 5.26 cm/yr presented in this paper in my opinion, especially when accounting for the underestimation in LOS values, as the authors dis-

cussed in the last paragraph of Section 4.2. Besides, the three rock glaciers in Janke et al. (2005) cannot represent "most other North American rock glaciers". The authors may consider changing their conclusions or drawing different comparisons.

15. Line 295–296, are there any references supporting this alternative explanation proposed here? Some studies suggest a contrasting point of view that the rock glacier accelerates when ice content decreases (Arenson et al., 2002), or a non-linear relationship between ice content and surface velocity (Cicoira et al., 2019).

Arenson, L., Hoelzle, M., & Springman, S. (2002, Apr-Jun). Borehole deformation measurements and internal structure of some rock glaciers in Switzerland. Permafrost and Periglacial Processes, 13(2), 117-135. https://doi.org/10.1002/ppp.414

Cicoira, A., Beutel, J., Faillettaz, J., Gartner-Roer, I., & Vieli, A. (2019, Mar). Resolving the influence of temperature forcing through heat conduction on rock glacier dynamics: a numerical modeling approach. Cryosphere, 13(3), 927-942. https://doi.org/10.5194/tc-13-927-2019

16. Line 308–316, this part is not discussing rock glacier velocity. Please consider restructuring this section.

17. Figure 8, Line 758–759, this sentence is unclear to me. Please explain how to scale the ascending and descending LOS and the purpose of that.

18. Line 418–421, I would suggest the authors specify those rock glacier velocities are InSAR-derived LOS velocities, otherwise the readers may misinterpret them as 3D creep velocities.

Technical corrections

1. Line 81 and 106, these two parts are better to be numbered as 2 and 3, as there is no Section 2 in this manuscript, and I don't see clear relations between the two subsections "Study area" and "InSAR analysis".

---

## Referee Comment (RC2) · Anonymous Referee #2 · 28 Jan 2021

This paper presents a new inventory of 255 active rock glaciers in the Uinta Mountains, Utah, from velocity maps of InSAR. The authors compared their inventory to the previous studies and discussed several aspects of the datasets, including the geomorphic and dynamic patterns, temporal displacements on the selected three rock glaciers, possible responses to climate changes, and the hydrological implications. This study shows the strength of InSAR for mapping and investigating active rock glaciers, although it is not the first one. The study also gives insights into how the unique climate change pattern in Uinta Mountains, which is different from the other places like European Alps and Asian Himalaya, would influence the dynamics of the rock glaciers there. The paper is overall well written and structured and can be accepted after minor revisions. My concerns mainly lay in the methodology part. Some details of the data processing need to be clarified or explained.

(1) The uncertainties of the surface velocities from InSAR should be evaluated. Significantly, the centimeter-level magnitude of velocities of rock glaciers presented here should be carefully interpreted because the atmospheric errors always reach such magnitude. The author asserted that they use the ERA-I global weather model to mitigate tropospheric delay in Sentinel-1 interferograms. However, the correction performance of the low-resolution ERA-I data may degrade at the small-scale targets like rock glaciers.

(2) The author compared their InSAR-based inventory to the inventory of Munroe et al. (2018), whose inventory method should be also summarized in the paper. Munroe et al. (2018) may compile both the active and inactive rock glaciers, while this study only compiles the active ones.

(3) The sensitivity of InSAR LOS measurements vary with respect to the aspects of rock glaciers. This may explain why little correlations were found between the InSAR LOS measurements and the topo-climate factors. The authors may calculate surface velocities along the downslope directions of rock glaciers and then probe the correlations.

**Specific comments**

Line 81 Add sub-title for section 2, e.g., '2 Study area and InSAR analysis.'

Line 91 Does 'Average precipitation' refer to the mean annual precipitation?

Line 125 The author stated that "To improve spatial resolution, selected one-year interferogram pairs were reprocessed with a USGS 3DEP DEM with 10 m pixel spacing". Which year of the image pairs were selected? Also, if the high-resolution DEM with 10 m spacing is available, why did the author remove the topographic phase using the SRTM data that has a coarser resolution (~ 30 m).

Line 146 Please elaborate on how did you address the average annual velocities from the ascending and descending stacks of 1-year interferograms since the observations from ascending and descending

SAR data have different looking directions. Furthermore, from my understanding, should average annual velocities be improved by averaging three-year InSAR observations, rather than only using the 1-year data.

Line 160 Please indicate the local reference points for phase unwrapping in Fig. 3 for the three selected rock glaciers.

Line 100 Please give a short summary of the inventory method used by Munroe et al., (2018), and the method for estimating the storage water of the rock glaciers.

Line 217 Rock glacier velocities cannot be correlated with 'morphology.'

Line 282 LOS velocity is a projection of real ground 3D velocity along the Satellite side-looking direction. It seems arbitrary by simply saying 'LOS measurements underestimate the true 3D velocity'.

Line 300. Please note that the correlation analysis between surface velocities and topo-climate factors requires that the surface velocities are in the same direction. The non-correlation pattern may also arise due to the diverse aspects of the rock glaciers.

Line 315 The statistical differences between this study and Munroe et al. (2018) may also be a result of the two studies' different inventorying methods. Munroe's (2018) inventory consists of both active and inactive rock glaciers, while this study only includes the active ones.

Line 374 The presence of 155 inactive rock glaciers supports this claim.

**Comments on Figures**

Figure 5. Add captions for Fig. 5c.

Figure 8. More displacement time series points are expected to be shown as 26 ascending, and 32 descending SAR scenes have been used to perform the SAR time series analysis. In comparison, it seems that no more than 20 displacement points are shown in (a-c).

---

## Author Comment (AC1) · 16 Feb 2021

**Response to Reviewer #2**

**Overall Comments**

This paper presents a new inventory of 255 active rock glaciers in the Uinta Mountains, Utah, from velocity maps of InSAR. The authors compared their inventory to the previous studies and discussed several aspects of the datasets, including the geomorphic and dynamic patterns, temporal displacements on the selected three rock glaciers, possible responses to climate changes, and the hydrological implications. This study shows the strength of InSAR for mapping and investigating active rock glaciers, although it is not the first one. The study also gives insights into how the unique climate change pattern in Uinta Mountains, which is different from the other places like European Alps and Asian Himalaya, would influence the dynamics of the rock glaciers there. The paper is overall well written and structured and can be accepted after minor revisions. My concerns mainly lay in the methodology part. Some details of the data processing need to be clarified or explained.

| No. | Comment | Response |
|---|---|---|
| 1 | *The uncertainties of the surface velocities from InSAR should be evaluated. Significantly, the centimeter-level magnitude of velocities of rock glaciers presented here should be carefully interpreted because the atmospheric errors always reach such magnitude. The author asserted that they use the ERA-I global weather model to mitigate tropospheric delay in Sentinel-1 interferograms. However, the correction performance of the low-resolution ERA-I data may degrade at the small-scale targets like rock glaciers.* | Concur. We examined the InSAR stacks carefully with and without the atmospheric correction and ensured the rock glacier deformation signals were consistent. In addition to using the TRAIN software package to reduce atmospheric InSAR noise, we mitigated atmospheric effects significantly by averaging multiple interferograms to create the stacks we used to calculate rock glacier velocities. Furthermore, we carefully selected local stable reference such as bedrock outcrops and parking lots which cancels out spatially correlated signals at distances exceeding the separation between these pixels.

To better explain our uncertainties, we will add a sentence or two evaluating atmospheric errors in detail to the paragraph in the discussion where we discuss the limitations of our methods (lines 307-326). We will also quantify the InSAR velocity of stable hillslopes throughout the Uintas and report this as the mean and standard deviation uncertainty. |
| 2 | *The author compared their InSAR-based inventory to the inventory of Munroe et al. (2018), whose inventory method should be also summarized in the paper. Munroe et al. (2018) may compile both the active and inactive rock glaciers, while this study only compiles the active ones.* | Concur. We will add a sentence to the introduction briefly summarizing the inventory method used in Munroe (2018) (Line 78). Munroe (2018) states:

"Rock glaciers were identified by scanning the bases of steep bedrock slopes and talus, searching for locations where the normal smooth talus profile is interrupted by a notably steep-fronted bulge with reduced lichen cover, where the talus appears wrinkled, and where |

| | | furrows and other evidence of movement are apparent. Areas exhibiting these characteristics were delineated as polygons in ArcMap GIS." |
|---|---|---|
| 3 | *The sensitivity of InSAR LOS measurements vary with respect to the aspects of rock glaciers. This may explain why little correlations were found between the InSAR LOS measurements and the topo-climate factors. The authors may calculate surface velocities along the downslope directions of rock glaciers and then probe the correlations.* | We did consider projecting rock glacier motion along the downslope direction. We decided against it because 1) we would be required to assume the rock glaciers were moving exactly in the steepest downslope direction based on DEMs made decades ago, which could introduce significant error, and 2) we don't think it would ultimately reveal a relationship between rock glacier velocity and topo-climatic factors. As far as we are aware, rock glacier velocity is often not well correlated topo-climatic factors because velocity depends on many variables (ice content, pore pressure, thickness, etc…). We further note that there is no relationship in our data between aspect and elevation, slope, rock glacier area, precipitation, or (with our low-resolution temperature data) temperature. |

**Specific Comments**

| No. | Comment | Response |
|---|---|---|
| 1 | *Line 81 Add sub-title for section 2, e.g., '2 Study area and InSAR analysis.'* | As per reviewer #1's suggestion, we plan on splitting this section into Section 2: Study area and Section 3: InSAR analysis. |
| 2 | *Line 91 Does 'Average precipitation' refer to the mean annual precipitation?* | Yes! We will revise the sentence to read:

"Mean annual precipitation (MAP) in the Uintas between 1981 and 2010 ranged from 45 to 107 cm (Fig. 2c)" |
| 3 | *Line 125 The author stated that "To improve spatial resolution, selected one-year interferogram pairs were reprocessed with a USGS 3DEP DEM with 10 m pixel spacing". Which year of the image pairs were selected? Also, if the high-resolution DEM with 10 m spacing is available, why did the author remove the topographic phase using the SRTM data that has a coarser resolution (~ 30 m).* | Our ascending stack included interferograms: 20160921 20170922 20160921 20170910 20160927 20170922 20170805 20180731 20180731 20190807

Our descending stack included interferograms: 20160902 20170828 20160902 20170909 20160926 20170921 20170804 20180730 20180730 20190806

We can include these lists of the interferograms in each stack in the appendix. |

| | | Computational limitations prevented us from processing all interferograms with the 10 m DEM. Instead, we used a 30 m DEM initially, then reprocessed our best interferograms with the 10-m DEM. Section will be revised to read: "To improve spatial resolution, selected one-year interferogram pairs were reprocessed with a USGS 3DEP DEM with 10 m pixel spacing. Computational limitations prevented us from processing all interferograms with the 10 m DEM." |
|---|---|---|
| 4 | *Line 146 Please elaborate on how did you address the average annual velocities from the ascending and descending stacks of 1-year interferograms since the observations from ascending and descending SAR data have different looking directions. Furthermore, from my understanding, should average annual velocities be improved by averaging three-year InSAR observations, rather than only using the 1-year data.* | We calculated 75th percentile LOS velocity for each rock glacier using both stacks. The larger of the ascending and descending values is used to represent rock glacier velocity in our data analysis (line 153-154). We avoided processing 3-year pairs, in part because we wanted to avoid unwrapping errors. See line 320: very long-baseline interferograms would be likely to introduce inaccuracies. |
| 5 | *Line 160 Please indicate the local reference points for phase unwrapping in Fig. 3 for the three selected rock glaciers.* | Concur. We'll add reference points to Fig. 3. |
| 6 | *Line 100 Please give a short summary of the inventory method used by Munroe et al., (2018), and the method for estimating the storage water of the rock glaciers.* | Concur. See response to general comment #2. See lines 394-394 for a brief summary of the method for estimating water content used by Munroe, (2018). |
| 7 | *Line 217 Rock glacier velocities cannot be correlated with 'morphology.'* | By morphology, we're referring to whether the rock glacier is tongue-shaped or lobate. To be more clear, we will edit this line to read: "No metric of rock glacier velocity is significantly correlated with rock glacier area, elevation, slope, aspect, or rock glacier type (Fig. 7a, Fig. A2)." |
| 8 | *Line 282 LOS velocity is a projection of real ground 3D velocity along the Satellite side-looking direction. It seems arbitrary by simply saying 'LOS measurements underestimate the true 3D velocity'.* | Since rock glacier motion is never entirely along the look direction, LOS velocity will, in practice, always be an underestimate of the rock glaciers' true 3d surface motion. We think this is important to mention, since it partly explains why our velocity estimates are low. |
| 9 | *Line 300. Please note that the correlation analysis between surface velocities and topo-climate factors requires that the surface velocities are in the same direction. The non-correlation pattern may also arise due to the diverse aspects of the rock glaciers.* | Characteristic rock glacier velocities were calculated by taking the 75th percent value of the velocity values within a rock glacier body. Two values were generated for each rock glacier, one derived from the ascending and another from the descending stack. The larger |

| | | (in terms of magnitude) of the ascending and descending values is used to represent rock glacier velocity in our data analysis. Before attempting to correlate rock glacier velocity estimates with topo-climatic factors, we also took their absolute value. While our estimates for rock glacier velocity are derived from different LOSs, this shouldn't have much effect on our ability to correlate topo-climatic factors with velocity, since we are considering only the magnitude of rock glacier motion here. |
|---|---|---|
| 10 | *Line 315 The statistical differences between this study and Munroe et al. (2018) may also be a result of the two studies' different inventorying methods. Munroe's (2018) inventory consists of both active and inactive rock glaciers, while this study only includes the active ones.* | It is possible that in reality there are relatively more north-facing inactive rock glaciers than inactive rock glaciers facing other directions, which would explain why the Munroe (2018) inventory, which contains active and inactive rock glaciers, has a higher proportion of north-facing rock glaciers than our active rock glacier inventory. However, we don't have a reasonable explanation for why more inactive rock glaciers would face north than other directions. It seems more likely that we simply underestimated the number of north-facing active rock glaciers due to InSAR's insensitivity to motion along the azimuth direction. |
| 11 | *Line 374 The presence of 155 inactive rock glaciers supports this claim.* | Looks like the comment here may be missing? |

**Figure Comments**

| 1 | *Figure 5. Add captions for Fig. 5c.* | Concur. We'll add a sentence that reads:

"(c) aspect of steep slopes (>10˚) of the Uinta Mountains, for reference." |
|---|---|---|
| 2 | *Figure 8. More displacement time series points are expected to be shown as 26 ascending, and 32 descending SAR scenes have been used to perform the SAR time series analysis. In comparison, it seems that no more than 20 displacement points are shown in (a-c).* | See line 159-160: "Interferograms with low overall coherence were manually removed from the time series." For clarity, we'll add a sentence to the same effect to the Figure 8 caption. |

Thank you very much for your comments!

---

## Author Comment (AC2) · 16 Feb 2021

**Response to Reviewer #1**

**Overall Comments**

The authors used InSAR technique to map and characterize rock glacier movement in a region where previous knowledge of rock glaciers is limited. It produces a new dataset that sheds light on the kinematic behavior of those permafrost landforms and provides interesting insights as to how the rock glaciers respond to the climatic conditions and their potential local hydrological importance in the future. Hopefully, this paper will be published and help generate more interest in studying rock glaciers in North America.

It is a well-written paper in general. However, I would raise a few issues mostly regarding the necessary details of the InSAR method adopted in this work. Accuracy in terminology and clarity in the argument can be further improved in a few places. Please see my comments below.

| No. | Comment | Response |
|---|---|---|
| 1 | *Line 21: the definition of rock glaciers here is inaccurate because they are not entirely "perennially frozen bodies", the upper part of which is seasonally frozen ground or the so-called active layer.* | Concur. Sentence will be simplified to read:

 "Rock glaciers are bodies of ice and rock debris that creep downslope due to deformation of their internal ice-rock mixture." |
| 2 | *Line 32–34, it might be inappropriate to draw an analogy between rock glaciers and ice glaciers here, because some of the enumerated drivers (e.g., liquid water, pore water pressure) influence the motion of the two types of landforms in ways that can hardly be regarded as similar.* | Here we are only referring the first-order relationships between changes in pore pressure and deformation of ice glaciers, faults, and landslides. We added some additional references to help clarify:

 "As with ice glaciers (e.g., Bartholomew et al., 2010; Iverson, 2010; Minchew and Meyer (2020) tectonic faults (e.g., Bürgmann, 2018) and landslides (e.g., Bayer et al,. 2018 Handwerger et al., 2019), liquid water, and pore-water pressure are also important drivers of short-term rock glacier motion (Ikeda et al., 2008; Moore, 2014; Kenner et al., 2017; Eriksen et al., 2016; Cicoira, 2019; Fey and Krainer, 2020)." |
| 3 | *Line 46, shear horizon is NOT "at the base of the rock glacier". Borehole investigations have revealed that sediments exist below the shear horizon, though the motion of which is negligible. The authors may refer to the two papers cited in the caption of Figure 1(i.e., Arenson et al., 2002; Kenner et al., 2017) and modify Figure 1b and 1d accordingly.* | Concur. Arenson et al. (2002) states that shear horizons can be located at different depths within a rock glacier. Sentence will be made more general to read:

 "Recent work suggests that spring acceleration is driven by water infiltrating shear horizons within rock glaciers, increasing pore-water pressure and reducing frictional strength (Kenner et al., 2017; Cocoira et al., 2019; Fey and Krainer, 2020)."

 Figure 1 will be modified, moving the shear horizon up slightly to reflect that the shear |

| | | zone is not necessarily located near the base of the rock glacier |
|---|---|---|
| 4 | *Line 69–70, what are the "significant patterns"?* | The sentence refers to significant patterns in rock glacier kinematics across the Uintas revealed by InSAR. The sentence will be made more explicit as follows:

"Many of these can be mitigated by careful study design, however, and at the scale of range-wide analysis, significant patterns in rock glacier kinematics can still be identified." |
| 5 | *Line 124–125, why do the authors use the 10-m resolution DEM for selected one-year pairs only, instead of applying it to all data?* | Computational limitations prevented us from processing all interferograms with the 10 m DEM. Instead, we used a 30 m DEM initially, then reprocessed our best interferograms with the 10-m DEM. Section will be revised to read:

"To improve spatial resolution, selected one-year interferogram pairs were reprocessed with a USGS 3DEP DEM with 10 m pixel spacing. Computational limitations prevented us from processing all interferograms with the 10 m DEM." |
| 6 | *Line 138–139, the description "LOS velocity signal consistent with the downslope direction" is confusing, because a LOS signal is obviously always in the LOS direction, which is from the ground to the satellite, and thus cannot be consistent with downslope direction.* | The LOS velocity signal can be negative or positive, indicating displacement toward or away from the satellite. For us to classify a rock glacier as active, the direction of the LOS signal indicated by the sign must suggest significant downslope movement. For clarity, the sentence will be revised to read:

"Rock glaciers displaying a clear and relatively high LOS velocity signal with a sign suggesting downslope movement were considered active (Fig. 2)." |
| 7 | *Line 147, which one-year interferograms do the authors use for calculating annual velocities? Here the authors mention both ascending and descending stacks of interferograms, however, Figure 2 only shows results derived from one descending track.* | Our ascending stack included interferograms:
20160921-20170922
20160921-20170910
20160927-20170922
20170805-20180731
20180731-20190807

Our descending stack included interferograms:
20160902-20170828
20160902-20170909
20160926-20170921
20170804-20180730
20180730-20190806 |

| | | These selected one-year pairs showed the lowest level of atmospheric effects. We calculated 75th percentile LOS velocity for each rock glacier using both stacks. The larger of the ascending and descending values is used to represent rock glacier velocity in our data analysis (line 153-154).

We will highlight these interferograms (red for ascending, blue for descending) in Table A1 to indicate that they were used to construct our interferogram stacks. |
|---|---|---|
| 8 | *Line 149, why do the authors remove negative LOS values? The motion towards the satellite is possible and Figure 2a does include negative values.* | We apologize as our language here is confusing. We didn't remove negative values- we took their magnitude in order to make all displacement values positive. This made it more straightforward to determine average LOS velocities over the surface of each rock glacier without negative and positive values cancelling each other out, leading to average LOS velocities with erroneously low magnitudes.

The sentence that reads:

"We use the velocity magnitude to remove negative LOS values that are caused by motion towards the satellite."

will be removed, as the prior sentence states:

"Average LOS velocity magnitudes were calculated by taking the mean of the absolute value of velocity values over the surface of each rock glacier." |
| 9 | *In Figure 2a, Line 715–716, the authors should specify the time span they used to calculate the average velocity, instead of just providing satellite orbital information. Also, the legend shows the unit of the velocity map in distance unit (cm) which may confuse the readers. Is Figure 2a a displacement map or a velocity map?* | Concur. In the caption we will mention that Figure 2a shows our average descending one-year pair stack. The figure is a velocity map, and the legend should be changed to read: "velocity, (cm/yr)" |
| 10 | *In Figure 2a, Line 715, the legend shows the unit of the velocity map is in centimeters which may confuse the readers. And the period of the observation should be specified.* | Concur. See response to comment #9. |

| 11 | *Line 163–172, this part is not under the topic of "InSAR analysis". The authors may consider reorganizing the structure of this section. Please also refer to the first technical correction below.* | Concur. We will add a new subsection header above line 163 titled, "2.3 Climate Data." |
|---|---|---|
| 12 | *Line 236–239 and Figure 2, Line 716, the previous inventory (Munroe, 2018) didn't classify the mapped rock glaciers based on their activities. How do the authors identify the inactive rock glaciers from the previously published dataset? If the inactive rock glaciers are landforms that do not show displacement in the interferograms, is it possible that some of those landforms are actually active, but their activity is not detected by InSAR, due to limitations of the technique, such as decorrelation, shadow, overlay, or the flow direction of landform is insensitive to the LOS direction?* | That's correct, we classified rock glaciers from the previous inventory that don't show displacement in the interferograms as inactive (line 204-205). We will edit the second sentence beginning on line 716 to read:

"Black polygons represent rock glaciers identified in the previous inventory (Munroe, 2018) which are inactive (i.e., show no active deformation) in our InSAR velocity maps."

In addition, we will add a sentence to our methods section beginning on Line 140 that states:

"Rock glaciers identified in the previous inventory that showed no deformation in our InSAR velocity maps were classified as inactive."

It is possible that some rock glaciers are active, but their activity was hidden by 1) decorrelation, 2) InSAR geometry,  3) the flow direction of the landform being insensitive to the LOS direction, or 4) motion at rates less than a few mm per year. I'll address each possibility. 1) In general, decorrelation over the rock glacier surface was very infrequent in our one-year stack velocity maps. 2) There were some instances where rock glaciers were partially or mostly hidden by InSAR geometry, but this was uncommon. 3) Rock glaciers tended to have multiple directions of flow, and we used interferograms derived from two tracks with different look directions. It's possible that rock glaciers flowing directly orthogonal to the satellite look direction, could have appeared inactive. 4) Rock glaciers moving slower than a few mm per year may be considered essentially inactive.

It's certainly possible that an active rock glacier appeared inactive in our velocity maps, however, it seems unlikely that this would be a widespread issue for 155 rock glaciers. We should acknowledge that this is a possibility. |

| | | We will add a sentence to that effect into the paragraph in the discussion addressing limitations of our methods, which begins on line 307. |
|---|---|---|
| 13 | *Line 279–280, the references here do not fully fit. Delaloye et al. (2010) focus on the Swiss Alps which is a regional study and cannot represent rock glaciers "around the world".* | Concur. We will edit this sentence to read:

"This range of mean velocities is lower than velocities reported for other rock glaciers in the western US and the Alps (Janke et al. 2005, Delaloye et al., 2010)." |
| 14 | *Line 283–285, Janke et al. (2005) reported average velocities of 7.3, 6.3, and 9.5 cm/yr for three rock glaciers in the Front Range, which are not notably faster than the LOS rates between 0.88 and 5.26 cm/yr presented in this paper in my opinion, especially when accounting for the underestimation in LOS values, as the authors discussed in the last paragraph of Section 4.2. Besides, the three rock glaciers in Janke et al. (2005) cannot represent "most other North American rock glaciers". The authors may consider changing their conclusions or drawing different comparisons.* | It is correct that the average velocities of the three rock glaciers reported by Janke are not notably faster than the rock glaciers in the Uintas. However, Table 1 in Janke et al. (2005) compiles velocity measurements of other North American rock glaciers, most of which have velocities above 10 cm/yr, and several of which have velocities above 50 cm/yr. For clarity, the reference in line 283-285 will be changed to read, "(Table 1 in Janke et al., 2005)" |
| 15 | *Line 295–296, are there any references supporting this alternative explanation proposed here? Some studies suggest a contrasting point of view that the rock glacier accelerates when ice content decreases (Arenson et al., 2002), or a non-linear relationship between ice content and surface velocity (Cicoira et al., 2019).*

*Arenson, L., Hoelzle, M., & Springman, S. (2002, Apr-Jun). Borehole deformation measurements and internal structure of some rock glaciers in Switzerland. Permafrost and Periglacial Processes, 13(2), 117-135. https://doi.org/10.1002/ppp.414*

*Cicoira, A., Beutel, J., Faillettaz, J., Gartner-Roer, I., & Vieli, A. (2019, Mar). Resolving the influence of temperature forcing through heat conduction on rock glacier dynamics: a numerical modeling approach. Cryosphere, 13(3), 927-942. https://doi.org/10.5194/tc-13-927-2019* | To our knowledge, there are no references supporting this alternative explanation. We agree that rock glaciers could theoretically accelerate with decreased ice content, and/or there could be a non-linear relationship between velocity and ice content. However, as evinced by the presence of relict rock glaciers, at some critically small ice/debris ratio, rock glaciers must decelerate. We suggest that many of the rock glaciers in the Uintas may have a small enough quantity of ice to cause deceleration. This threshold may be higher in the Uintas than other places due to dryer conditions and increasing aridity, minimizing liquid water in the rock glacier body. |
| 16 | *Line 308–316, this part is not discussing rock glacier velocity. Please consider restructuring this section.* | Concur. We will create a new subsection called "Limitations of InSAR" |

| 17 | *Figure 8, Line 758–759, this sentence is unclear to me. Please explain how to scale the ascending and descending LOS and the purpose of that.* | The time series results provide cumulative displacement in the LOS direction. As such, cumulative displacement can be negative and decreasing or positive and increasing (motion towards or away from the satellite). In this figure, we took the absolute value of displacement so to make all displacement positive and increasing for ease of comparison between rock glaciers. |
|----|----|----|
| 18 | *Line 418–421, I would suggest the authors specify those rock glacier velocities are InSAR-derived LOS velocities, otherwise the readers may misinterpret them as 3D creep velocities.* | Concur. We will revise the caption accordingly. |

**Technical Corrections**

| 1 | *Line 81 and 106, these two parts are better to be numbered as 2 and 3, as there is no Section 2 in this manuscript, and I don't see clear relations between the two subsections "Study area" and "InSAR analysis".* | Concur. We'll number these sections 2 and 3. |
|----|----|----|

Thank you very much for providing comments!

---

## Referee Comment (RC3) · Anonymous Referee #3 · 30 Mar 2021

In this manuscript the authors used satellite SAR interferometry to identify and monitor active rock glaciers in the Uinta Mountains (Utah, USA). Velocity maps derived from Sentinel-1 data were considered to generate an inventory of active rock glaciers. A number of relationships with topographic and climatic drivers were calculated and analyesd. Mean LOS velocities are in the order of a few cm/yr. The paper is very well structured and written. However, there are some important missing information that should be included in a revised version.

l.    10.    According    to    the    ongoing    work    of    the    IPA
Action    Group:    Rock    glacier    inventories    and    kinematics

(https://www.unifr.ch/geo/geomorphology/en/research/ipa-action-group-rock-glacier) regarding the definition of standard guidelines for inventorying rock glaciers (https://bigweb.unifr.ch/Science/Geosciences/Geomorphology/Pub/Website/IPA/Guidelines/V4/200507_Baseline_Concep the following updated categorization of activity are proposed: - An active rock glacier shows coherent downslope movement over most of its surface. As an indication, the displacement rate can range from a decimeter to several meters per year. - Transitional rock glacier shows little to no downslope movement over most of its surface. As an indication, the average displacement rate is less than a decimeter per year in an annual mean over most of the rock glacier. Downslope movement must not be confused with subsidence. The rock glaciers in the study area seem thus to be rather transitional and not active.

l. 25-26. Also the other way round is valid: rock glaciers might be considered as indicators of climate change, see again the work of the IPA Action Group: Rock glacier inventories and kinematics and in particular the Task 2 activities "Rock glacier kinematics as an associated parameter of ECV Permafrost", https://bigweb.unifr.ch/Science/Geosciences/Geomorphology/Pub/Website/IPA/RGK/200121_RockGlacierKinematics_V1.

l. 65-75. Add further references, e.g.: Strozzi et al. Detecting and quantifying mountain permafrost creep from in situ inventory, space-borne radar interferometry and airborne digital photogrammetry. Int. J. Rem. Sens. 2004, 25, 2919–2931. Barboux et al. Inventorying slope movements in an Alpine environment using DinSAR. Earth Surf. Process. Landf. 2014, 39, 2087–2099. Rick et al. Detection and inventorying of slope movements in the Brooks Range, Alaska using DInSAR: A test study. In Proceedings of the GEOQuébec 2015: 68th Canadian Geotechnical Conference and 7th Canadian Permafrost Conference, Quebec City Convention Centre, Québec, QC, Canada, 20–23 September 2015. Necsoiu et al. Rock glacier dynamics in Southern Carpathian Mountains from high-resolution optical and multi-temporal SAR satellite imagery. Remote Sens. Environ. 2016, 177, 21–36. Strozzi et al. Monitoring Rock Glacier Kinematics with Satellite Synthetic Aperture Radar, Remote Sens. 2020, 12(3), 559.

l. 125. Why only selected one-year pairs and not all?

l. 136-145. This methodological part is not well explained: - What do you mean at l. 136 with "InSAR velocity maps"? One ascending and one descending? Or for all the InSAR pairs analyzed (see Table in the appendix)? - What do you mean by "a clear and relatively high LOS velocity signal"? Be more precise and quantitative. - See IPA guidelines for the definition of the activity classes (first point above). - What do you mean by "delineated"? Manually or automatically?

l. 146-148. What is the difference between these "average annual velocities" and those of the previous section? How were these maps computed? Which pairs were considered? They could be highlighted in the table of the appendix. Any weighting (e.g. time interval, coherence) in the average?

What is shown in Figures 2 and 3? The velocities of l. 136-145 or those of l. 146-148?

l. 204. A threshold for inactive rock glaciers was not defined. Please be precise, considering also the indications of the IPA working group.

l. 207. What is the min. detectable size of an InSAR signal?

l. 212-214. Add a reference to these statements.

l. 219-221. As observed in other regions, please add appropriate references.

l. 236-239 and l. 295-298. Again, better define what is an active rock glacier, in particular considering the recent IPA guidelines. In this region we are probably at the limit of permafrost occurrence, small activity is possibly linked to the presence of permafrost.

l. 320. Why were these apparently wrong estimates (40 cm/a in 12 days versus 4 cm/a in 1 year) not masked out?

l. 333. . . . and else where, add references.

---

## Author Comment (AC4) · 28 Apr 2021

**Response to Reviewer #3**

**Overall Comments**

In this manuscript the authors used satellite SAR interferometry to identify and monitor active rock glaciers in the Uinta Mountains (Utah, USA). Velocity maps derived from Sentinel-1 data were considered to generate an inventory of active rock glaciers. A number of relationships with topographic and climatic drivers were calculated and analyzed. Mean LOS velocities are in the order of a few cm/yr. The paper is very well structured and written. However, there are some important missing information that should be included in a revised version.

| No. | Comment | Response |
|-----|---------|----------|
| 1 | *Line 10: According to the ongoing work of the IPA Action Group: Rock glacier inventories and kinematics (https://www.unifr.ch/geo/geomorphology/en/research/ipa-action-group-rock-glacier) regarding the definition of standard guidelines for inventorying rock glaciers (https://bigweb.unifr.ch/Science/Geosciences/Geomorphology/Pub/Website/IPA/Guidelines/V4/200507_Baseline_Concepts_) the following updated categorization of activity are proposed: - An active rock glacier shows coherent downslope movement over most of its surface. As an indication, the displacement rate can range from a decimeter to several meters per year. - Transitional rock glacier shows little to no downslope movement over most of its surface. As an indication, the average displacement rate is less than a decimeter per year in an annual mean over most of the rock glacier. Downslope movement must not be confused with subsidence. The rock glaciers in the study area seem thus to be rather transitional and not active.* | Concur. According to these proposed definitions, most of the actively creeping rock glaciers we identified can be categorized as transitional by virtue of their slow velocities.

We will add a sentence to Line 30 of our introduction:

"Rock glaciers with slow movement (<10 cm/yr) only detectable by measurement and/or restricted to areas of non-dominant extent have been defined as transitional and evolve towards an active on inactive state according to their topographic and climatic setting (IPA, 2020)"

We will revise our language throughout the paper to refer to rock glaciers moving at <10 cm/yr as transitional. |
| 2 | *Line 25-26: Also the other way round is valid: rock glaciers might be considered as indicators of climate change, see again the work of the IPA Action Group: Rock glacier inventories and kinematics and in particular the Task 2 activities "Rock glacier kinematics as an associated parameter of ECV Permafrost", https://bigweb.unifr.ch/Science/Geosciences/Geomorphology/Pub/Website/IPA/RGK/200121_RockGlacierKinematics_V1.0.* | Concur. |
| 3 | *Line 65-75: Add further references, e.g.: Strozzi et al. Detecting and quantifying mountain* | Concur. We will add these references. |

| | | |
|---|---|---|
| | *permafrost creep from in situ inventory, space-borne radar interferometry and airborne*
*digital photogrammetry. Int. J. Rem. Sens. 2004, 25, 2919–2931.*

*Barboux et al. Inventorying slope movements in an Alpine environment using DinSAR. Earth Surf. Process. Landf. 2014, 39, 2087–2099.*

*Rick et al. Detection and inventorying of slope movements in the Brooks Range, Alaska using DInSAR: A test study. In Proceedings of the GEOQuébec 2015: 68th Canadian Geotechnical Conference and 7th Canadian Permafrost Conference, Quebec City Convention Centre, Québec, QC, Canada, 20–23 September 2015.*

*Necsoiu et al. Rock glacier dynamics in Southern Carpathian Mountains from high-resolution optical and multi-temporal SAR satellite imagery. Remote Sens. Environ. 2016, 177, 21–36. Strozzi et al. Monitoring Rock Glacier Kinematics with Satellite Synthetic Aperture Radar, Remote Sens. 2020, 12(3), 559.* | |
| 4 | *Line 125: Why only selected one-year pairs and not all?* | Computational limitations prevented us from processing all interferograms with the 10 m DEM. Instead, we used a 30 m DEM initially, then reprocessed our best interferograms with the 10-m DEM. Section will be revised to read:

"To improve spatial resolution, selected one-year interferogram pairs were reprocessed with a USGS 3DEP DEM with 10 m pixel spacing. Computational limitations prevented us from processing all interferograms with the 10 m DEM." |
| 5 | *Line 136-145: This methodological part is not well explained: - What do you mean at l. 136 with "InSAR velocity maps"? One ascending and one descending? Or for all the InSAR pairs analyzed (see Table in the appendix)? - What do you mean by "a clear and relatively high LOS velocity signal"? Be more precise and quantitative. - See IPA guidelines for the definition of the activity classes (first point above). - What do you mean by "delineated"? Manually or automatically?* | We used velocity maps derived from all the analyzed InSAR pairs to generate our inventory. We typically relied on one-year pairs more, as displacement signals in one-year pairs were much larger than any signals related to atmospheric noise. However, in the case of fast-moving rock glaciers that may have caused decorrelation errors in one-year pairs, shorter baseline interferograms were frequently used as well. |

By "clear and relatively high LOS velocity signal" we mean that rock glaciers obviously displacing at a faster rate than their surroundings were considered active. We did not use a specific velocity threshold to determine whether rock glaciers were active. As long as pixels over the surface of the mapped rock glacier body showed a clear and consistent displacement signal in a direction consistent with the downslope direction, and the surrounding pixels did not, we considered the rock glacier to be active.

We manually delineated rock glacier boundaries in QGIS.

We will revise this section to read:

"All resulting InSAR velocity maps were used along with Google Earth imagery, the USGS 10 m DEM, and the previous Uinta rock glacier inventory (Munroe, 2018) to generate an active rock glacier inventory in QGIS 3.10. Rock glaciers displaying a clear and relatively high LOS velocity signal with a sign suggesting downslope movement were considered active or transitional (Fig. 2). Boundaries of rock glaciers were manually delineated on the basis of morphology and InSAR-derived movement pattern. Slope, aspect, and elevation of features in the rock glacier inventory were calculated in QGIS from the 10 m DEM. Rock glaciers were classified as lobate or tongue-shaped (Barsch, 1996) based on morphology, and as "North Uintas" or "South Uintas" based on their location relative to the east-west trending spine of the mountain range (Fig. 2). A non-parametric Kruskall-Wallis test was used to establish significance of differences between groups."

| 6 | *146-148. What is the difference between these "average annual velocities" and those of the previous section? How were these maps computed? Which pairs were considered? They could be highlighted in the table of the appendix. Any weighting (e.g. time interval, coherence) in the average?* | These average annual velocities were calculated from stacks made from the velocity maps mentioned in the previous section (Line 136). These stacks (one ascending and one descending) were computed by averaging 1-year pair velocity maps, ignoring "NoData" values. |
| --- | --- | --- |
| | *What is shown in Figures 2 and 3? The velocities of l. 136-145 or those of l. 146-148?* | Our ascending stack included interferograms: 20160921 20170922 |

| | | 20160921 20170910 |
| | | 20160927 20170922 |
| | | 20170805 20180731 |
| | | 20180731 20190807 |
| | | |
| | | Our descending stack included interferograms: |
| | | 20160902 20170828 |
| | | 20160902 20170909 |
| | | 20160926 20170921 |
| | | 20170804 20180730 |
| | | 20180730 20190806 |
| | | |
| | | We will include these lists of the interferograms in each stack in the appendix (Table A2). |
| | | |
| | | There was no weighting in the average. All interferograms used were one-year pairs. We used a coherence threshold of 0.3 during interferogram processing to remove low-quality data. The "No Data" values produced as a result were ignored when averaging velocity maps to create the stacks. |
| | | |
| | | Lines 146-147 will be revised to read: |
| | | |
| | | "Average annual velocities for rock glaciers were calculated in QGIS using velocity maps derived from ascending and descending stacks of 1 year interferograms (Fig. 2). These stacks were calculated from the one-year pairs with 10 m pixel spacing (Table A2). Average LOS velocity magnitudes were calculated by taking the mean of the absolute value of velocity values over the surface of each rock glacier." |
| | | |
| | | Figures 2 and 3 both show the InSAR velocity map stacks. The caption for Figure 2 will be revised to read: |
| | | |
| | | "Uinta Mountains study site. (a) Hillshade map of the Uinta Mountains overlaid with InSAR average velocity stack from descending track 27. |
| | | |
| | | The caption for Figure 3 already includes that average velocity stacks were used in the figure. |
| 7 | *Line 204: A threshold for inactive rock glaciers was not defined. Please be precise, considering also the indications of the IPA working group.* | We did not use a specific velocity threshold to identify inactive rock glaciers. When the pixels over the surface of a mapped rock glacier body did not show clear and coherent displacement |

| | | visually distinct from the displacement of the surrounding pixels, the rock glacier was considered inactive. In practice, our slowest "active" rock glaciers move at rates >0.9 cm/yr, very close to the 1 cm/yr threshold used by the IPA action group to separate inactive rock glaciers from transitional rock glacier.

We will add a sentence to our methods section beginning on Line 140 that states: "Rock glaciers identified in the previous inventory that showed no coherent and distinct deformation in our InSAR velocity maps were classified as inactive." |
|---|---|---|
| 8 | *Line 207: What is the min. detectable size of an InSAR signal?* | InSAR can be used to accurately estimate displacement down to the millimeter scale. See:

Bürgmann, R., Rosen, P. A., & Fielding, E. J. (2000). Synthetic aperture radar interferometry to measure Earth's surface topography and its deformation. Annual review of earth and planetary sciences, 28(1), 169-209.

The smallest spatial area we considered to have a clear and coherent signal indicating rock glacier activity was 5,000 $m^2$. We will add a sentence to that effect to Line 140. |
| 9 | *Line 212-214: Add a reference to these statements.* | These are our own observations of Uinta rock glaciers.

For clarity, we will revise Line 212 to read:

"We observed that Uinta rock glaciers generally have…" |
| 10 | *Line 219-221: As observed in other regions, please add appropriate references.* | We included references for seasonal changes in rock glacier motion observed in other regions in our introduction, Lines 42-43. |
| 11 | *Lines 236-239 and 295-298: Again, better define what is an active rock glacier, in particular considering the recent IPA guidelines. In this region we are probably at the limit of permafrost occurrence, small activity is possibly linked to the presence of permafrost.* | See response to Comment 1. We will adopt the language of the IPA action group throughout the document. |
| 12 | *Line 320: Why were these apparently wrong estimates (40 cm/a in 12 days versus 4 cm/a in 1 year) not masked out?* | 1) There is no conclusive evidence that movement of this particular rock glacier caused unwrapping errors. It is possible that these discrepancies could be the result of particularly strong seasonal changes in |

| | | velocity. We didn't feel that removing this data was justified based on the evidence that we had.

 2) These apparent errors only appeared to impact a very small number of rock glaciers in our inventory (<5). They are unlikely to have a large impact on our velocity estimates. |
|---|---|---|
| **13** | *Line 333: ... and else where, add references* | Concur. We will revise the sentence to read:

 It's likely that this observation period was too small to capture possible long-term trends in rock glacier motion, as have been well-documented in the Alps and other regions (Delaloye et al., 2008; Kääb et al., 2007; Kaufmann and Ladstädter, 2007; Roer et al., 2005; Vonder Muehll et al., 2007; Eriksen et al., 2018; Necsoiu et al., 2016). |
| Thank you very much for providing comments! | | |

---

## Author Response (AR1)

**Author's Response to Comments**

**Reviewer 1**
**Overall Comments**
*The authors used InSAR technique to map and characterize rock glacier movement in a region where previous knowledge of rock glaciers is limited. It produces a new dataset that sheds light on the kinematic behavior of those permafrost landforms and provides interesting insights as to how the rock glaciers respond to the climatic conditions and their potential local hydrological importance in the future. Hopefully, this paper will be published and help generate more interest in studying rock glaciers in North America.*

*It is a well-written paper in general. However, I would raise a few issues mostly regarding the necessary details of the InSAR method adopted in this work. Accuracy in terminology and clarity in the argument can be further improved in a few places. Please see my comments below.*

| No. | Comment | Response |
|-----|---------|----------|
| 1 | *Line 21: the definition of rock glaciers here is inaccurate because they are not entirely "perennially frozen bodies", the upper part of which is seasonally frozen ground or the so-called active layer.* | Concur. Sentence has been simplified to read:

"Rock glaciers are bodies of ice and rock debris that creep downslope due to deformation of their internal ice-rock mixture." |
| 2 | *Line 32–34, it might be inappropriate to draw an analogy between rock glaciers and ice glaciers here, because some of the enumerated drivers (e.g., liquid water, pore water pressure) influence the motion of the two types of landforms in ways that can hardly be regarded as similar.* | Here we are only referring the first-order relationships between changes in pore pressure and deformation of ice glaciers, faults, and landslides. We added some additional references to help clarify (Lines 35-39):

"As with ice glaciers (e.g. Bartholomew et al., 2010; Iverson, 2010; Minchew and Meyer, 2020), tectonic faults (e.g. Bürgmann, 2018) and landslides (e.g. Bayer et al., 2018; Handwerger et al., 2019), liquid water and pore-water pressure are also important drivers of short-term rock glacier motion (Ikeda et al., 2008; Moore, 2014; Kenner et al., 2017; Eriksen et al., 2016; Cicoira, 2019; Fey and Krainer, 2020)." |
| 3 | *Line 46, shear horizon is NOT "at the base of the rock glacier". Borehole investigations have revealed that sediments exist below the shear horizon, though the motion of which is negligible. The authors may refer to the two papers cited in the caption of Figure 1(i.e., Arenson et al., 2002; Kenner et al., 2017) and modify Figure 1b and 1d accordingly.* | Concur. Arenson et al. (2002) states that shear horizons can be located at different depths within a rock glacier. The sentence has been made more general and now reads (Lines 50-52):

"Recent work suggests that spring acceleration is driven by melt water infiltration that increases pore-water pressure and reduces frictional strength along shear horizons within rock glaciers (Kenner et al., 2017; Cocoira et al., 2019; Fey and Krainer, 2020)." |

| | | Figure 1 has been modified to move the shear horizon up slightly, reflecting that the shear zone is not necessarily located near the base of the rock glacier. The Figure 1c caption has been modified to read:

"Plot showing how total displacement is related to shearing and plastic deformation in rock glaciers. Deformation primarily occurs along shear horizons within rock glaciers." |
|---|---|---|
| 4 | *Line 69–70, what are the "significant patterns"?* | The sentence refers to significant patterns in rock glacier kinematics across the Uintas revealed by InSAR. The sentence has been made more explicit as follows (Lines 73-75):

"Many of these can be mitigated by careful study design, however, and at the scale of range-wide analysis, significant patterns in rock glacier kinematics can still be identified." |
| 5 | *Line 124–125, why do the authors use the 10-m resolution DEM for selected one-year pairs only, instead of applying it to all data?* | Computational limitations prevented us from processing all interferograms with the 10 m DEM. Undergraduate student George Brencher only had access to a 2011 iMac with 4 GB of RAM and 500 GB of storage space. Instead, we used a 30 m DEM initially, then reprocessed our best interferograms with the 10 m DEM. The section has been revised to read (Lines 144-147):

"In addition, selected one-year interferogram pairs were reprocessed with a USGS 3DEP DEM with 10 m pixel spacing. The primary reason we reprocessed these selected pairs was to improve spatial resolution in order to more accurately inventory the moving rock glaciers (Table A1). However, computational limitations prevented us from processing all 108 interferograms with the 10 m DEM." |
| 6 | *Line 138–139, the description "LOS velocity signal consistent with the downslope direction" is confusing, because a LOS signal is obviously always in the LOS direction, which is from the ground to the satellite, and thus cannot be consistent with downslope direction.* | The LOS velocity signal can be negative or positive, indicating displacement toward or away from the satellite. For us to classify a rock glacier as moving, the direction of the LOS signal indicated by the sign must suggest significant downslope movement. For clarity, the sentence has been revised to read (Lines 160-161):

"We inventoried rock glaciers displaying a clear and relatively high LOS velocity signal with a sign suggesting downslope movement (Fig. 2)." |

| 7 | *Line 147, which one-year interferograms do the authors use for calculating annual velocities? Here the authors mention both ascending and descending stacks of interferograms, however, Figure 2 only shows results derived from one descending track.* | Our ascending stack included interferograms:
20160921-20170922
20160921-20170910
20160927-20170922
20170805-20180731
20180731-20190807

Our descending stack included interferograms:
20160902-20170828
20160902-20170909
20160926-20170921
20170804-20180730
20180730-20190806

These selected one-year pairs showed the lowest level of atmospheric effects. We calculated 75th percentile LOS velocity and downslope velocity for each rock glacier using both stacks. The larger of the ascending and descending downslope values is used to represent rock glacier velocity in our data analysis (Lines 189-190).

We have highlighted these interferograms (red for ascending, blue for descending) in Table A1 to indicate that they were used to construct our interferogram stacks.

The caption of Table A1 has been edited to read:

"Table A1. Track, date, and time span of all interferograms generated. Pairs that were averaged to create the ascending and descending one-year stacks used to estimate LOS velocities of rock glaciers are highlighted in red and blue, respectively." |
| --- | --- | --- |
| 8 | *Line 149, why do the authors remove negative LOS values? The motion towards the satellite is possible and Figure 2a does include negative values.* | We apologize as our language here was confusing. We didn't remove negative values- we took their magnitude in order to make all displacement values positive. This made it more straightforward to determine average LOS velocities over the surface of each rock glacier without negative and positive values cancelling each other out, leading to average LOS velocities with erroneously low magnitudes.

The sentence that reads: |

| | | |
|---|---|---|
| | | "We use the velocity magnitude to remove negative LOS values that are caused by motion towards the satellite."

has been removed, as the prior sentence (Lines 173-174) states:

"Average LOS velocity magnitudes were calculated by taking the mean of the absolute value of velocity values over the surface of each rock glacier." |
| 9 | *In Figure 2a, Line 715–716, the authors should specify the time span they used to calculate the average velocity, instead of just providing satellite orbital information. Also, the legend shows the unit of the velocity map in distance unit (cm) which may confuse the readers. Is Figure 2a a displacement map or a velocity map?* | Concur. The caption for 2a has been revised to read:

"(a) Hillshade map of the Uinta Mountains overlaid with average one-year InSAR velocity map derived from descending track 27."

Figure 2a is a velocity map. The legend has been changed to read:

"velocity (cm/yr)" |
| 10 | *In Figure 2a, Line 715, the legend shows the unit of the velocity map is in centimeters which may confuse the readers. And the period of the observation should be specified.* | Concur. See response to Comment #9. |
| 11 | *Line 163–172, this part is not under the topic of "InSAR analysis". The authors may consider reorganizing the structure of this section. Please also refer to the first technical correction below.* | Concur. We have added a new subsection header above Line 198 titled, "2.3 Climate Data." |
| 12 | *Line 236–239 and Figure 2, Line 716, the previous inventory (Munroe, 2018) didn't classify the mapped rock glaciers based on their activities. How do the authors identify the inactive rock glaciers from the previously published dataset? If the inactive rock glaciers are landforms that do not show displacement in the interferograms, is it possible that some of those landforms are actually active, but their activity is not detected by InSAR, due to limitations of the technique, such as decorrelation, shadow, overlay, or the flow direction of landform is insensitive to the LOS direction?* | That's correct; we classified rock glaciers from the previous inventory that don't show displacement in the interferograms as inactive (Lines 239-240). We have edited the second sentence beginning on Line 807 to read:

"Black polygons represent rock glaciers identified in the previous inventory (Munroe, 2018) which are inactive (i.e., show no coherent and distinct deformation) in our InSAR velocity maps."

In addition, we have added a sentence to our methods section beginning on Line 162 that states: |

| | | "Rock glaciers identified in the Munroe (2018) inventory that showed no coherent and distinct deformation in our InSAR velocity maps were classified as inactive." |
|---|---|---|
| | | It is possible that some rock glaciers are active, but their activity was hidden by 1) decorrelation, 2) InSAR geometry, 3) the flow direction of the landform being insensitive to the LOS direction, or 4) motion at rates less than a few mm per year. I'll address each possibility. 1) In general, decorrelation over the rock glacier surface was very infrequent in our one-year stack velocity maps. 2) There were some instances where rock glaciers were partially or mostly hidden by InSAR geometry, but this was uncommon. 3) Rock glaciers tended to have multiple directions of flow, and we used interferograms derived from two tracks with different look directions. It's possible that rock glaciers flowing directly orthogonal to the satellite look direction, could have appeared inactive. 4) Rock glaciers moving slower than a few mm per year may be considered essentially inactive. |
| | | It's certainly possible that an active rock glacier appeared inactive in our velocity maps, however, it seems unlikely that this would be a widespread issue for 205 rock glaciers. To address this possibility, we have added the sentence: |
| | | "Some rock glacier movement could also have been hidden by InSAR decorrelation or geometry; however, since these issues were quite uncommon in our velocity maps, they are not likely to have produced widespread inaccuracies or systematic bias in our inventory." |
| | | to line 358 in our discussion of the limitations of InSAR in our study. |
| 13 | *Line 279–280, the references here do not fully fit. Delaloye et al. (2010) focus on the Swiss Alps which is a regional study and cannot represent rock glaciers "around the world".* | Concur. We have edited this sentence to read (Lines 320-321): "This range of mean velocities is lower than velocities reported for other rock glaciers in the western US and the Alps (Janke et al. 2005, Delaloye et al., 2010)." |

| 14 | *Line 283–285, Janke et al. (2005) reported average velocities of 7.3, 6.3, and 9.5 cm/yr for three rock glaciers in the Front Range, which are not notably faster than the LOS rates between 0.88 and 5.26 cm/yr presented in this paper in my opinion, especially when accounting for the underestimation in LOS values, as the authors discussed in the last paragraph of Section 4.2. Besides, the three rock glaciers in Janke et al. (2005) cannot represent "most other North American rock glaciers". The authors may consider changing their conclusions or drawing different comparisons.* | It is correct that the average velocities of the three rock glaciers reported by Janke are not notably faster than the rock glaciers in the Uintas. However, Table 1 in Janke et al. (2005) compiles velocity measurements of other North American rock glaciers, most of which have velocities above 10 cm/yr, and several of which have velocities above 50 cm/yr. For clarity, the reference in Line 325 has been changed to read:

"(Table 1 in Janke et al., 2005)" |
|---|---|---|
| 15 | *Line 295–296, are there any references supporting this alternative explanation proposed here? Some studies suggest a contrasting point of view that the rock glacier accelerates when ice content decreases (Arenson et al., 2002), or a non-linear relationship between ice content and surface velocity (Cicoira et al., 2019).*

*Arenson, L., Hoelzle, M., & Springman, S. (2002, Apr-Jun). Borehole deformation measurements and internal structure of some rock glaciers in Switzerland. Permafrost and Periglacial Processes, 13(2), 117-135. https://doi.org/10.1002/ppp.414*

*Cicoira, A., Beutel, J., Faillettaz, J., Gartner-Roer, I., & Vieli, A. (2019, Mar). Resolving the influence of temperature forcing through heat conduction on rock glacier dynamics: a numerical modeling approach. Cryosphere, 13(3), 927-942. https://doi.org/10.5194/tc-13-927-2019* | To our knowledge, there are no references supporting this alternative explanation. We agree that rock glaciers could theoretically accelerate with decreased ice content, and/or there could be a non-linear relationship between velocity and ice content. However, as evinced by the presence of relict rock glaciers, at some critically small ice/debris ratio, rock glaciers must decelerate. We suggest that many of the rock glaciers in the Uintas may have a small enough quantity of ice to cause deceleration. This threshold may be higher in the Uintas than other places due to dryer conditions and increasing aridity, minimizing liquid water in the rock glacier body. |
| 16 | *Line 308–316, this part is not discussing rock glacier velocity. Please consider restructuring this section.* | Concur. We have created a new subsection called "Limitations of InSAR and Uncertainty Analysis" |
| 17 | *Figure 8, Line 758–759, this sentence is unclear to me. Please explain how to scale the ascending and descending LOS and the purpose of that.* | The time series results provide cumulative displacement in the LOS direction. As such, cumulative displacement can be negative and decreasing or positive and increasing (motion towards or away from the satellite). In this figure, we took the absolute value of displacement so as to make all displacement positive and increasing for ease of comparison between rock glaciers. |

| 18 | *Line 418–421, I would suggest the authors specify those rock glacier velocities are InSAR-derived LOS velocities, otherwise the readers may misinterpret them as 3D creep velocities.* | Concur. We now use downslope velocities as the characteristic velocity of the rock glaciers. Lines 480-483 have been revised to read:

"InSAR-derived downslope rock glacier velocities are between 0.35 and 6.03 cm/yr, with an average of 1.92 cm/yr, and are not correlated with variables including elevation, area, aspect, slope, or morphology. Time series analysis of three rock glaciers revealed a seasonal rhythm in LOS velocities, with an average of 4.42 cm/yr during the snow-free late summer, and 0.86 cm/yr during the rest of the year." |

**Technical Corrections**

| 19 | *Line 81 and 106, these two parts are better to be numbered as 2 and 3, as there is no Section 2 in this manuscript, and I don't see clear relations between the two subsections "Study area" and "InSAR analysis".* | Concur. We have numbered these sections 2 and 3 and added the header:

"3 Methods"

above the InSAR Analysis section. |

**Reviewer 2**
**Overall Comments**
*This paper presents a new inventory of 255 active rock glaciers in the Uinta Mountains, Utah, from velocity maps of InSAR. The authors compared their inventory to the previous studies and discussed several aspects of the datasets, including the geomorphic and dynamic patterns, temporal displacements on the selected three rock glaciers, possible responses to climate changes, and the hydrological implications. This study shows the strength of InSAR for mapping and investigating active rock glaciers, although it is not the first one. The study also gives insights into how the unique climate change pattern in Uinta Mountains, which is different from the other places like European Alps and Asian Himalaya, would influence the dynamics of the rock glaciers there. The paper is overall well written and structured and can be accepted after minor revisions. My concerns mainly lay in the methodology part. Some details of the data processing need to be clarified or explained.*

| No. | Comment | Response |
|---|---|---|
| 1 | *The uncertainties of the surface velocities from InSAR should be evaluated. Significantly, the centimeter-level magnitude of velocities of rock glaciers presented here should be carefully interpreted because the atmospheric errors always reach such magnitude. The author asserted that they use the ERA-I global weather model to mitigate tropospheric delay in Sentinel-1 interferograms. However, the correction performance of the low-resolution* | Concur. We examined the InSAR stacks carefully with and without the atmospheric correction and ensured the rock glacier deformation signals were consistent. In addition to using the TRAIN software package to reduce atmospheric InSAR noise, we mitigated atmospheric effects significantly by averaging multiple interferograms to create the stacks we used to calculate rock glacier velocities. Furthermore, we carefully selected local stable reference such as bedrock outcrops |

*ERA-I data may degrade at the small-scale targets like rock glaciers.*

and parking lots which cancels out spatially correlated signals at distances exceeding the separation between these pixels.

We have quantified uncertainty to some extent by estimating the average velocity of 12 stable reference areas in the Uintas using our one-year stacks. The following sentence has been added to our methods section (Lines 180-181):

"To quantify uncertainty, InSAR-derived LOS velocities for 12 mostly flat, roughly rock-glacier sized control areas were calculated using the absolute value of the same ascending and descending stacks used to calculate rock glacier velocity."

A corresponding paragraph has been added to results section 4.4, Rock Glacier Velocity (Lines 257-259):

"We approximated uncertainty in the InSAR velocity by quantifying the apparent velocity of 12 stable control areas. We found that average LOS velocities for stable areas was $0.33 \pm 0.12$ cm/yr and $0.62 \pm 0.32$ cm/yr for ascending and descending stacks respectively."

In addition, the following paragraph evaluating tropospheric error has been added to our new "Limitations of InSAR and Uncertainty Analysis" section (Lines 381-389).

"Additional uncertainty in our velocity estimates comes from tropospheric phase delay. We mitigated these errors by 1) estimating velocity using one-year pairs, which results in interferograms with a greater signal-to-noise ratio, 2) averaging multiple high-quality interferograms together and using the stack to estimate velocity, 3) implementing a tropospheric phase delay correction, and 4) choosing local stable reference points. However, due to the large areal extent, large altitudinal range, and highly variable topography of the study region, some non-negligible element of phase delay remains in the one year-stacks we used to estimate velocity (Fig. 2). In general, the descending stack suffered slightly more from atmospheric

| | | errors. The uncertainty in the InSAR velocities is unlikely to be systematic in nature as areas of positive and negative velocities appear to be randomly distributed at the elevations where rock glaciers occur. Importantly, the clearly and coherently moving areas of all active or transitional rock glaciers are moving above this level of uncertainty." |
|---|---|---|
| 2 | *The author compared their InSAR-based inventory to the inventory of Munroe et al. (2018), whose inventory method should be also summarized in the paper. Munroe et al. (2018) may compile both the active and inactive rock glaciers, while this study only compiles the active ones.* | Concur. We have added the following sentence to the introduction briefly summarizing the inventory method used in Munroe (2018) (Lines 95-98):

"The method of rock glacier identification employed in a previous study by Munroe (2018) involved examining the bases of high-angle bedrock and talus slopes for steep-fronted bulges with reduced lichen cover and ridges and furrows characteristic of rock glaciers using visual imagery at a 1:5,000 scale." |
| 3 | *The sensitivity of InSAR LOS measurements vary with respect to the aspects of rock glaciers. This may explain why little correlations were found between the InSAR LOS measurements and the topo-climate factors. The authors may calculate surface velocities along the downslope directions of rock glaciers and then probe the correlations.* | Concur. We calculated downslope velocities for our inventory using the method described in Liu et al. 2013. The following text has been added to the methods section (Lines 184-190)

"In addition to our LOS velocity estimates, we calculated ground surface velocity estimates by projecting our LOS estimates onto the downslope direction using the method described in Liu et al. (2013). This approach assumes that rock glaciers are flowing uniformly along their average azimuth and slope direction. The heading angle and incidence angle of the satellite are ~12˚ (positive counter clockwise from North) and ~41˚ for the ascending track and ~168˚ and ~40˚ for the descending track. The USGS 10 m DEM was used to calculate average slope angle and average azimuth angle for each rock glacier and the 75th percentile LOS velocities were used to represent LOS velocity. The larger of the resulting ascending and descending values is used to represent rock glacier velocity in our analysis."

We found no correlations between downslope velocities and topo-climatic factors other than aspect. Rock glacier with north and south-facing average aspects had higher downslope velocities (Figure A2). This is very likely due |

| | | to aggressive scaling of velocities of rock glaciers with north and south facing average aspects that deform in multiple directions in reality. |
|---|---|---|

**Specific Comments**

| No. | Comment | Response |
|---|---|---|
| 3 | *Line 81 Add sub-title for section 2, e.g., '2 Study area and InSAR analysis.'* | As per reviewer #1's suggestion, we have split this section into Section 2: Study area and Section 3: Methods, 3.1, InSAR Analysis. |
| 4 | *Line 91 Does 'Average precipitation' refer to the mean annual precipitation?* | Yes, we have revised the sentence (Lines 111-112) to read:

"Mean annual precipitation (MAP) in the Uintas between 1981 and 2010 ranged from 45 to 107 cm (Fig. 2c)." |
| 5 | *Line 125 The author stated that "To improve spatial resolution, selected one-year interferogram pairs were reprocessed with a USGS 3DEP DEM with 10 m pixel spacing". Which year of the image pairs were selected? Also, if the high-resolution DEM with 10 m spacing is available, why did the author remove the topographic phase using the SRTM data that has a coarser resolution (~ 30 m).* | Our ascending stack included interferograms:
20160921 20170922
20160921 20170910
20160927 20170922
20170805 20180731
20180731 20190807

Our descending stack included interferograms:
20160902 20170828
20160902 20170909
20160926 20170921
20170804 20180730
20180730 20190806

We have highlighted these interferograms (red for ascending, blue for descending) in Table A1 to indicate that they were used to construct our interferogram stacks.

The caption of Table A1 has been edited to read:

"Table A1. Track, date, and time span of all interferograms generated. Pairs that were averaged to create the ascending and descending one-year stacks used to estimate LOS velocities of rock glaciers are highlighted in red and blue, respectively."

Computational limitations prevented us from processing all interferograms with the 10 m DEM. Undergraduate student George Brencher only had access to a 2011 iMac with 4 GB of |

| | | RAM and 500 GB of storage space. Instead, we used a 30 m DEM initially, then reprocessed our best interferograms with the 10-m DEM. This section (Lines 144-147) has been revised to read:

"In addition, selected one-year interferogram pairs were reprocessed with a USGS 3DEP DEM with 10 m pixel spacing. The primary reason we reprocessed these selected pairs was to improve spatial resolution in order to more accurately inventory the moving rock glaciers (Table A1). However, computational limitations prevented us from processing all 108 interferograms with the 10 m DEM." |
|---|---|---|
| 6 | *Line 146 Please elaborate on how did you address the average annual velocities from the ascending and descending stacks of 1-year interferograms since the observations from ascending and descending SAR data have different looking directions. Furthermore, from my understanding, should average annual velocities be improved by averaging three-year InSAR observations, rather than only using the 1-year data.* | We calculated 75th percentile LOS velocity for each rock glacier using both ascending and descending stacks. These values were used to calculate ascending and descending downslope velocities. The larger of the ascending and descending downslope values is used to represent rock glacier velocity in our data analysis.

We avoided processing 3-year pairs, in part because we wanted to avoid unwrapping errors. See Line 372: very long-baseline interferograms would be likely to introduce inaccuracies. |
| 7 | *Line 160 Please indicate the local reference points for phase unwrapping in Fig. 3 for the three selected rock glaciers.* | Concur. We have added the reference points to Fig. 3. A sentence has been added to the caption of Fig. 3 (Lines 820-821) that reads:

"Yellow squares represent stable local reference points used in our time series analysis." |
| 8 | *Line 100 Please give a short summary of the inventory method used by Munroe et al., (2018), and the method for estimating the storage water of the rock glaciers.* | Concur. See response to general comment #2. See lines 457-458 for a brief summary of the method for estimating water content used by Munroe, (2018). |
| 9 | *Line 217 Rock glacier velocities cannot be correlated with 'morphology.'* | By morphology, we're referring to whether the rock glacier is tongue-shaped or lobate. To be more clear, we have edited this sentence (Lines 254-255) to read:

"No metric of rock glacier velocity is significantly correlated with rock glacier area, elevation, slope, aspect, or type (i.e. lobate or tongue-shaped) (Fig. 7a, Fig. A2)." |

| 10 | *Line 282 LOS velocity is a projection of real ground 3D velocity along the Satellite side-looking direction. It seems arbitrary by simply saying 'LOS measurements underestimate the true 3D velocity'.* | Since rock glacier motion is never entirely along the look direction, LOS velocity will, in practice, always be an underestimate of the rock glaciers' true 3d surface motion. We think this is important to mention, since it partly explains why our LOS velocity estimates are low. |
|---|---|---|
| 11 | *Line 300. Please note that the correlation analysis between surface velocities and topo-climate factors requires that the surface velocities are in the same direction. The non-correlation pattern may also arise due to the diverse aspects of the rock glaciers.* | Concur. Considering values derived from only the ascending or only the descending stack, there was still no correlation between rock glacier LOS velocity and topo-climate variables. We have revised Figure A2 to include separate ascending stack and descending stack velocity estimates for each rock glacier in order to demonstrate this. In addition, we have added the following sentence to the Fig. A2 caption:

"Two velocity values are presented for each rock glacier; one is derived from the ascending one-year stack (red) and one from the descending one-year stack (blue)." |
| 12 | *Line 315 The statistical differences between this study and Munroe et al. (2018) may also be a result of the two studies' different inventorying methods. Munroe's (2018) inventory consists of both active and inactive rock glaciers, while this study only includes the active ones.* | After revising our active and inactive rock glacier inventories, we no longer find a notable difference in the proportion of rock glaciers facing north. |
| 13 | *Line 374 The presence of 155 inactive rock glaciers supports this claim.* | Looks like the comment here may be missing. |

**Figure Comments**

| 14 | *Figure 5. Add captions for Fig. 5c.* | Concur. We have added a caption that reads:

"(c) Aspect of steep slopes (>10˚) of the Uinta Mountains, for reference." |
|---|---|---|
| 15 | *Figure 8. More displacement time series points are expected to be shown as 26 ascending, and 32 descending SAR scenes have been used to perform the SAR time series analysis. In comparison, it seems that no more than 20 displacement points are shown in (a-c).* | See Line 195: "Interferograms with low overall coherence were manually removed from the time series." For clarity, we have added a sentence to the Figure 8 caption that reads:

"Interferograms with low overall coherence were manually removed from the time series." |

**Reviewer 3**
**Overall Comments**
*In this manuscript the authors used satellite SAR interferometry to identify and monitor active rock glaciers in the Uinta Mountains (Utah, USA). Velocity maps derived from Sentinel-1 data were*

*considered to generate an inventory of active rock glaciers. A number of relationships with topographic and climatic drivers were calculated and analyzed. Mean LOS velocities are in the order of a few cm/yr. The paper is very well structured and written. However, there are some important missing information that should be included in a revised version.*

| No. | Comment | Response |
|---|---|---|
| 1 | *Line 10: According to the ongoing work of the IPA Action Group: Rock glacier inventories and kinematics (https://www.unifr.ch/geo/geomorphology/en/research/ipa-action-group-rock-glacier) regarding the definition of standard guidelines for inventorying rock glaciers (https://bigweb.unifr.ch/Science/Geosciences/Geomorphology/Pub/Website/IPA/Guidelines/V4/200507_Baseline_Concepts_) the following updated categorization of activity are proposed: - An active rock glacier shows coherent downslope movement over most of its surface. As an indication, the displacement rate can range from a decimeter to several meters per year. - Transitional rock glacier shows little to no downslope movement over most of its surface. As an indication, the average displacement rate is less than a decimeter per year in an annual mean over most of the rock glacier. Downslope movement must not be confused with subsidence. The rock glaciers in the study area seem thus to be rather transitional and not active.* | Concur. According to these proposed definitions, most of the actively creeping rock glaciers we identified can be categorized as transitional by virtue of their slow velocities.

We have added the following sentences to our introduction (Lines 29-33):

"Active rock glaciers contain internal ice, exhibit coherent downslope movement over most of their surface, and move downslope at rates on the order of a decimeter to several meters per year (IPA Action Group Rock Glacier Inventories and Kinematics, 2020). Rock glaciers moving at slower rates (<10 cm/yr) are defined as transitional and can evolve towards an active or inactive state depending on their topographic and climatic setting (IPA Action Group Rock Glacier Inventories and Kinematics, 2020)."

We have revised our language throughout the paper to refer to rock glaciers moving at <10 cm/yr as transitional.

We have also added the following citation to the IPA Baseline Concepts document:

"IPA Action Group Rock Glacier Inventories and Kinematics: Baseline Concepts Version 4.1. Université de Fribourg Geomorphology Research Group, https://www.unifr.ch/geo/geomorphology/en/research/ipa-action-group-rock-glacier/, 2020." |
| 2 | *Line 25-26: Also the other way round is valid: rock glaciers might be considered as indicators of climate change, see again the work of the IPA Action Group: Rock glacier inventories and kinematics and in particular the Task 2 activities "Rock glacier kinematics as an associated parameter of ECV Permafrost", https://bigweb.unifr.ch/Science/Geosciences/Geomorphology/Pub/Website/IPA/RGK/200121_RockGlacierKinematics_V1.0.* | Concur. |

| 3 | *Line 65-75: Add further references, e.g.: Strozzi et al. Detecting and quantifying mountain permafrost creep from in situ inventory, space-borne radar interferometry and airborne digital photogrammetry. Int. J. Rem. Sens. 2004, 25, 2919–2931.* | Concur. We have added these references. |
|---|---|---|
| | *Barboux et al. Inventorying slope movements in an Alpine environment using DinSAR. Earth Surf. Process. Landf. 2014, 39, 2087–2099.* | |
| | *Rick et al. Detection and inventorying of slope movements in the Brooks Range, Alaska using DInSAR: A test study. In Proceedings of the GEOQuébec 2015: 68th Canadian Geotechnical Conference and 7th Canadian Permafrost Conference, Quebec City Convention Centre, Québec, QC, Canada, 20–23 September 2015.* | |
| | *Necsoiu et al. Rock glacier dynamics in Southern Carpathian Mountains from high-resolution optical and multi-temporal SAR satellite imagery. Remote Sens. Environ. 2016, 177, 21–36. Strozzi et al. Monitoring Rock Glacier Kinematics with Satellite Synthetic Aperture Radar, Remote Sens. 2020, 12(3), 559.* | |
| 4 | *Line 125: Why only selected one-year pairs and not all?* | Computational limitations prevented us from processing all interferograms with the 10 m DEM. Undergraduate student George Brencher only had access to a 2011 iMac with 4 GB of RAM and 500 GB of storage space. Instead, we used a 30 m DEM initially, then reprocessed our best interferograms with the 10-m DEM. The section (Lines 144-147) has been revised to read: |
| | | "In addition, selected one-year interferogram pairs were reprocessed with a USGS 3DEP DEM with 10 m pixel spacing. The primary reason we reprocessed these selected pairs was to improve spatial resolution in order to more accurately inventory the moving rock glaciers (Table A1). However, computational limitations prevented us from processing all 108 interferograms with the 10 m DEM." |

| 5 | *Line 136-145: This methodological part is not well explained: - What do you mean at l. 136 with "InSAR velocity maps"? One ascending and one descending? Or for all the InSAR pairs analyzed (see Table in the appendix)? - What do you mean by "a clear and relatively high LOS velocity signal"? Be more precise and quantitative. - See IPA guidelines for the definition of the activity classes (first point above). - What do you mean by "delineated"? Manually or automatically?* | We used velocity maps derived from all the analyzed InSAR pairs to generate our inventory. We typically relied on one-year pairs more, as displacement signals in one-year pairs were much larger than any signals related to atmospheric noise. However, in the case of fast-moving rock glaciers that may have caused decorrelation errors in one-year pairs, shorter baseline interferograms were frequently used as well.

By "clear and relatively high LOS velocity signal" we mean that rock glaciers obviously displacing at a faster rate than their surroundings were considered to be moving. We did not use a specific velocity threshold to determine whether rock glaciers were moving. As long as pixels over the surface of the mapped rock glacier body showed a clear and consistent displacement signal in a direction consistent with the downslope direction, and the surrounding pixels did not, we considered the rock glacier to be moving.

We manually delineated rock glacier boundaries in QGIS.

We have revised this section to read:

"All resulting InSAR velocity maps were used along with Google Earth imagery, the USGS 10 m DEM, and the previous Uinta rock glacier inventory (Munroe, 2018) to generate a new rock glacier inventory in QGIS 3.10. We inventoried rock glaciers displaying a clear and relatively high LOS velocity signal with a sign suggesting downslope movement (Fig. 2). The smallest spatial area we considered to have a clear and coherent signal indicating rock glacier activity was 5,000 m2. Rock glaciers identified in the Munroe (2018) inventory that showed no coherent and distinct deformation in our InSAR velocity maps were classified as inactive. Boundaries of rock glaciers were manually delineated on the basis of morphology and InSAR-derived movement pattern. Slope, aspect, and elevation of features in the rock glacier inventory were calculated in QGIS from the 10 m DEM. Rock glaciers were classified as lobate (width > length) or tongue-shaped (length > width) (Barsch, 1996) based |

| | | |
|---|---|---|
| | | on morphology. We also grouped the rock glaciers into "North Uintas" or "South Uintas" based on their location relative to the east-west trending spine of the mountain range (Fig. 2). A non-parametric Kruskall-Wallis test with a significance threshold of 0.05 was used to establish significance of differences between all rock glacier groups." |
| **6** | *146-148. What is the difference between these "average annual velocities" and those of the previous section? How were these maps computed? Which pairs were considered? They could be highlighted in the table of the appendix. Any weighting (e.g. time interval, coherence) in the average?*

*What is shown in Figures 2 and 3? The velocities of l. 136-145 or those of l. 146-148?* | These average annual velocities were calculated from stacks made from the velocity maps mentioned in the previous section (Line 159). These stacks (one ascending and one descending) were computed by averaging 1-year pair velocity maps, ignoring "NoData" values.

Our ascending stack included interferograms:
20160921 20170922
20160921 20170910
20160927 20170922
20170805 20180731
20180731 20190807

Our descending stack included interferograms:
20160902 20170828
20160902 20170909
20160926 20170921
20170804 20180730
20180730 20190806

We have highlighted these interferograms (red for ascending, blue for descending) in Table A1 to indicate that they were used to construct our interferogram stacks.

The caption of Table A1 has been edited to read:

"Table A1. Track, date, and time span of all interferograms generated. Pairs that were averaged to create the ascending and descending one-year stacks used to estimate LOS velocities of rock glaciers are highlighted in red and blue, respectively."

There was no weighting in the average. All interferograms used were one-year pairs. We used a coherence threshold of 0.3 during interferogram processing to remove low-quality data. The "No Data" values produced |

| | | as a result were ignored when averaging velocity maps to create the stacks. |
|---|---|---|
| | | Lines 171-174 have been revised to read: |
| | | "Average annual velocities for rock glaciers were calculated in QGIS using velocity maps derived from ascending and descending stacks of one-year interferograms (Fig. 2). These stacks were calculated from the one-year pairs with 10 m pixel spacing (Table A1). Average LOS velocity magnitudes were calculated by taking the mean of the absolute value of velocity values over the surface of each rock glacier." |
| | | Figures 2 and 3 both show the InSAR velocity map stacks. The caption for Figure 2 has been revised to read: |
| | | "(a) Hillshade map of the Uinta Mountains overlaid with average one-year InSAR velocity map derived from descending track 27." |
| | | The caption for Figure 3 already includes that average velocity stacks were used in the figure. The bottom row of Figure 3 shows velocity, so the legend title has been revised to read "velocity," and entries are reported in cm/yr. |
| 7 | *Line 204: A threshold for inactive rock glaciers was not defined. Please be precise, considering also the indications of the IPA working group.* | We did not use a specific velocity threshold to identify inactive rock glaciers. When the pixels over the surface of a mapped rock glacier body did not show clear and coherent displacement visually distinct from the displacement of the surrounding pixels, the rock glacier was considered inactive. |
| | | We have added a sentence to our methods section beginning on Line 162 that states: |
| | | "Rock glaciers identified in the Munroe (2018) inventory that showed no coherent and distinct deformation in our InSAR velocity maps were classified as inactive." |
| 8 | *Line 207: What is the min. detectable size of an InSAR signal?* | InSAR can be used to accurately estimate displacement down to the millimeter scale. See: |
| | | Bürgmann, R., Rosen, P. A., & Fielding, E. J. (2000). Synthetic aperture radar interferometry to measure Earth's surface topography and its |

| | | deformation. Annual review of earth and planetary sciences, 28(1), 169-209. |
|---|---|---|
| | | In terms of minimum spatial area, we have added the following sentence to Line 161: |
| | | "The smallest spatial area we considered to have a clear and coherent signal indicating rock glacier activity was 5,000 $m^2$." |
| **9** | *Line 212-214: Add a reference to these statements.* | These are our own observations of Uinta rock glaciers. |
| | | For clarity, we have revised Line 248 to read: |
| | | "Uinta rock glaciers generally have non-uniform spatial velocity patterns." |
| **10** | *Line 219-221: As observed in other regions, please add appropriate references.* | We included references for seasonal changes in rock glacier motion observed in other regions in our introduction, Lines 47-50. |
| **11** | *Lines 236-239 and 295-298: Again, better define what is an active rock glacier, in particular considering the recent IPA guidelines. In this region we are probably at the limit of permafrost occurrence, small activity is possibly linked to the presence of permafrost.* | See response to Comment 1. We have adopted the language of the IPA rock glacier action group throughout the document. |
| **12** | *Line 320: Why were these apparently wrong estimates (40 cm/a in 12 days versus 4 cm/a in 1 year) not masked out?* | 1) There is no conclusive evidence that movement of this particular rock glacier caused unwrapping errors. It is possible that these discrepancies could be the result of particularly strong seasonal changes in velocity. We didn't feel that removing this data was justified based on the evidence that we had. 2) These apparent errors only appeared to impact a very small number of rock glaciers in our inventory (<5). They are unlikely to have a large impact on our velocity estimates. |
| **13** | *Line 333: … and else where, add references* | Concur. We have revised the sentence to read (Lines 394-396): |
| | | "It's likely that this observation period was too small to capture possible long-term trends in rock glacier motion, as have been well-documented in the Alps and other regions (Delaloye et al., 2010; Kääb et al., 2007; Kaufmann and Ladstädter, 2007; Roer et al., 2005; Vonder Muehll et al., 2007; Eriksen et al., 2018; Necsoiu et al., 2016)." |

**Editor Comments**

**Terminology Comments**

| No. | Comment | Revision |
|---|---|---|
| 1 | *Line 29: Make the definition of active rock glaciers less ambiguous: also inactive rock glaciers contain ice (if less) but do not move. So make clear that for an active rock glacier to be defined as such it contains ice AND it has to move at given velocities. Maybe: active rock glaciers are defined as… Or active rock glaciers move at velocities…* | Concur. We have adopted rock glacier activity categories of the IPA action group as per Reviewer 3's comments and have updated our definition of active rock glaciers accordingly. The sentence has been changed to read (Lines 29-31):

"Active rock glaciers contain internal ice, exhibit coherent downslope movement over most of their surface, and move downslope at rates on the order of a decimeter to several meters per year (IPA Action Group Rock Glacier Inventories and Kinematics, 2020)." |
| 2 | *Line 32: I would remove ice from line 32 in "ice glaciers" as it is an unusual term for "normal" glaciers. I would say there is agreement that when referring to glaciers we refer to what you call "ice" glaciers.* | While we agree that the term "glaciers" in general implies ice glaciers, we feel that in this context it is important to make a clear distinction between rock glaciers and ice glaciers. The term "ice glaciers" is unusual, but we think it adds necessary clarity to the text. |

**Literature Review and Motivations Comments**

| 3 | *The authors mention that InSAR has been used already to map rock glaciers, create rock glaciers inventories and study rock kinematics, and provide references for this, but no details of the success and limitations of these studies, and the actual findings of them. I would encourage them to add a short paragraph in the Introduction where they discuss what has been done to date, strengths and weaknesses of previous inSAR works and where they make clear where their contribution stands in this context.* | Concur. We have added a new paragraph as follows (Line 76-87):

"InSAR has been effectively used to create rock glacier inventories of varying extents and to study rock glacier kinematics (e.g. Nagler et al., 2002; Rignot et al., 2002; Kenyi and Kaufman, 2003; Strozzi et al., 2004; 2020; Lilleøren et al., 2013; Lui et al., 2013; Barboux et al., 2014; Rick et al., 2015; Necsoiu et al., 2016; Wang et al., 2017; Villarroel et al., 2018). Previous InSAR-aided studies of rock glaciers generally fall into two groups: 1) studies that leverage InSAR to facilitate inventorying of large numbers of rock glaciers (e.g. Villarroel et al. (2018) mapped 2,116 rock glaciers) over large regions (e.g. Wang et al. (2017) examined an area of 63,000 km$^2$) or 2) studies that apply InSAR for in-depth kinematic analysis of one or a few rock glaciers (e.g. Rignot et al. 2002; Kenyi and Kaufman, 2003; Necsoiu et al., 2016). We combine these two approaches by inventorying rock glaciers over an area of roughly 3,000 km$^2$, estimating velocity for all rock glaciers in |

| | | our inventory, and performing displacement time series analysis on three representative rock glaciers. Our interpretations of our inventory results and kinematic results inform one another, forming a more comprehensive picture of Uinta rock glacier characteristics and dynamics." |
|---|---|---|
| **4** | *The same lack of a more extended discussion of current literature is evident in parts also in the Discussion section. Also here it would be beneficial to compare their findings more extensively to those of previous studies: the authors do this when they compare velocities and arid/non arid environments but it could be more extensive and comprehensive (e.g. addressing also aspects, shapes, etc).* | Concur. Some consideration of additional literature was added to our discussion of aspect (Lines 306-307):

"Rock glaciers may preferentially face north due to decreased sunlight exposure on north-facing slopes, which decreases local temperatures (Munroe, 2018). This trend has been reported for other North American rock glaciers (e.g. Luckman and Crockett, 1978; Janke et al., 2007; Johnson et al., 2007; Liu et al., 2013)." |

**Section Structure and Numbering Comments**

| | | |
|---|---|---|
| **5** | *I miss an overarching section 2. After 1.Introduction the paper starts with 2.1. Study area, which is then followed by 2.2. InSAR analysis. A possible structure could be: 2.Methods, and then their order of sections; or 2_Study site and data and 3. Methods (including 3.1. InSAR analysis and 3.2. Classification of rock glaciers).*

*Indeed, I would encourage the authors to include a section where they explain how they determine the attributes of glaciers (e.g. lobate versus tongue shaped, etc). I might have missed that but do not think so.* | Concur. As per Reviewer 1's comments, we have numbered these sections 2 and 3 and added the header:

"3 Methods"

above the InSAR Analysis section.

We have revised Lines 166-168 in our methods to read:

"Slope, aspect, and elevation of features in the rock glacier inventory were calculated in QGIS from the 10 m DEM. Rock glaciers were classified as lobate (width > length) or tongue-shaped (length > width) (Barsch, 1996) based on morphology. We also grouped the rock glaciers into "North Uintas" or "South Uintas" based on their location relative to the east-west trending spine of the mountain range (Fig. 2)."

to make our classification of rock glaciers more clear. |

**Rock Glacier Inventory and Statistical Significance Comments**

| | | |
|---|---|---|
| **6** | *The authors discuss that lobate rock glaciers are found at a significantly lower elevation than tongue-shaped rock glaciers. Did you do* | Yes. All comparisons between rock glacier groups were evaluated using a non-parametric Kruskall-Wallis test. P values are listed in |

| | | |
|---|---|---|
| | *a significant test on the two populations? I would encourage the authors to test the statistical significance of their statement.*

*The same comment applies for all the other occurrences of significant: rock glaciers in the North being significantly higher than those in the south, or significant smaller. All those should be backed by a test for statistical significance.*

*From the caption of Figure 4 it seems they did test for statistical significance, but this information should be included in the text (in a Method section) and what they did should be clearly explained (including the level of statistical significance used).* | results section. A P value of 0.05 was used as a threshold for significance. We have revised Lines 168-169 in our methods to read:

"A non-parametric Kruskall-Wallis test with a significance threshold of 0.05 was used to establish significance of differences between all rock glacier groups." |

**Other Comments**

| | | |
|---|---|---|
| 7 | *Line 80: Make clear what is the multi-annual scale here (provide the years and/or period)* | Concur. We have edited the sentence beginning on Line 98 to read:

"In contrast, we use satellite-based InSAR alongside visual imagery to identify active and transitional rock glaciers and to evaluate controls on their rates of motion over seasonal to multi-annual time scales (2016-2019)." |
| 8 | *Figure 3: Top panels: it is difficult to clearly see both the shape and surface texture/morphology of the three selected glaciers. For Grayling in particular the 3D perspective is poor. I would suggest you enlarge the glacier in each panel by zooming in on them (especially for Whiterocks and Rockflour there is a lot of other area that is not needed but takes space), modify the view angle and/or choose a different depiction.*

*Add in the caption the period for which the displacement values in the bottom panels are calculated.* | Concur. The scale of the images on the top row has been shrunk and the view has been changed to overhead.

The legend in the bottom row has been revised such that the title is "velocity," and the entries are reported in centimeters per year.

The figure caption for the top row has been edited to read:

"(top) Overhead view of rock glaciers in © Google Earth, annotated in Adobe Illustrator." |
| 9 | *Figure 1 and 2: This is only a suggestion but I would swap the order of Figure 1 and 2. In Figure 1 the study site is already presented (panel a), and the authors do not discuss that figure extensively (only in the very first line of the introduction, line 22, for a statement tement before discussion of figure 2.* | The aim of Figure 1 is to reinforce and augment the introductory information we provide about rock glaciers in the paper's opening sentences, while Figure 2 is generally meant to accompany Section 2: Study Area. We don't feel that switching the order of the two would be helpful. |

| 10 | *DEMs used: on page 5 the authors say they use the 10m USGS DEM, but they have used the SRTM DEM (of 30m resolution) earlier in the processing of the interferograms to remove the topography signal and geocode them. Can they explain why two DEMs and why not using directly the USGS one of higher resolution?* | Computational limitations prevented us from processing all interferograms with the 10 m DEM. Undergraduate student George Brencher only had access to a 2011 iMac with 4 GB of RAM and 500 GB of storage space. Instead, we used a 30 m DEM initially, then reprocessed our best interferograms with the 10-m DEM. The section (Lines 144-147) has been revised to read:

"In addition, selected one-year interferogram pairs were reprocessed with a USGS 3DEP DEM with 10 m pixel spacing. The primary reason we reprocessed these selected pairs was to improve spatial resolution in order to more accurately inventory the moving rock glaciers (Table A1). However, computational limitations prevented us from processing all 108 interferograms with the 10 m DEM" |
|----|----|----|
| 11 | *Figures caption: some more detailed are needed here in general.* | Concur. As per the requests of the reviewers, the following additions have been made to the figure captions:

Lines 806-809 of the Figure 2a caption have been revised to read:

"(a) Hillshade map of the Uinta Mountains overlaid with average one-year InSAR velocity map derived from descending track 27. Red polygons represent active rock glaciers identified in this study. Black polygons represent rock glaciers identified in the previous inventory (Munroe, 2018) which are inactive (i.e., show no coherent and distinct deformation) in our InSAR velocity maps."

A caption has been added for Figure 5c:

"(c) Aspect of steep slopes (>10˚) of the Uinta Mountains, for reference."

A sentence has been added to the Figure 8a caption that reads:

"Interferograms with low overall coherence were manually removed from the time series."

A sentence has been added to the Table A1 caption that reads: |

| | | "Pairs that were averaged to create the ascending and descending one-year stacks used to estimate LOS velocities of rock glaciers are highlighted in red and blue, respectively." |
|---|---|---|
| **12** | *Uncertainty analysis: I understand the +/- values the authors provide with their estimates of e.g. velocities, displacements etc are the standard deviations over their population. Is it possible to provide an estimate of uncertainty associated with the InSAR analysis?* | Concur. We have added more information about the InSAR uncertainty to the manuscript. We quantified uncertainty by measuring the InSAR-derived LOS velocities for 12 flat, roughly rock-glacier sized control areas (Lines 180-182).

In the results section in Lines 257-259, we now state:

"We approximated uncertainty in the InSAR velocity by quantifying the apparent velocity of 12 stable control areas. We found that average LOS velocities for stable areas was $0.33 \pm 0.12$ cm/yr and $0.62 \pm 0.32$ cm/yr for ascending and descending stacks respectively."

In addition, we added a new section titled "Limitations of InSAR and Uncertainty Analysis," which contains the following paragraph evaluating InSAR uncertainty (Lines 362-370):

"Additional uncertainty in our velocity estimates comes from tropospheric phase delay. We mitigated these errors by 1) estimating velocity using one-year pairs, which results in interferograms with a greater signal-to-noise ratio, 2) averaging multiple high-quality interferograms together and using the stack to estimate velocity, 3) implementing a tropospheric phase delay correction, and 4) choosing local stable reference points. However, due to the large areal extent, large altitudinal range, and highly variable topography of the study region, some non-negligible element of phase delay remains in the one year-stacks we used to estimate velocity (Fig. 2). In general, the descending stack suffered slightly more from atmospheric errors. The uncertainty in the InSAR velocities is unlikely to be systematic in nature as areas of positive and negative velocities appear to be randomly distributed at the elevations where rock glaciers occur. Importantly, the clearly and coherently moving areas of all active or |

| | | transitional rock glaciers are moving above this level of uncertainty." |
|---|---|---|